# A model of hippocampal replay driven by experience and environmental structure facilitates spatial learning

Nicolas Diekmann[1,2], Sen Cheng[1,2]*

[1]Institute for Neural Computation, Faculty of Computer Science, Ruhr University Bochum, Bochum, Germany; [2]International Graduate School of Neuroscience, Ruhr University Bochum, Bochum, Germany

**Abstract** Replay of neuronal sequences in the hippocampus during resting states and sleep play an important role in learning and memory consolidation. Consistent with these functions, replay sequences have been shown to obey current spatial constraints. Nevertheless, replay does not necessarily reflect previous behavior and can construct never-experienced sequences. Here, we propose a stochastic replay mechanism that prioritizes experiences based on three variables: 1. Experience strength, 2. experience similarity, and 3. inhibition of return. Using this prioritized replay mechanism to train reinforcement learning agents leads to far better performance than using random replay. Its performance is close to the state-of-the-art, but computationally intensive, algorithm by Mattar & Daw (2018). Importantly, our model reproduces diverse types of replay because of the stochasticity of the replay mechanism and experience-dependent differences between the three variables. In conclusion, a unified replay mechanism generates diverse replay statistics and is efficient in driving spatial learning.

## Editor's evaluation

This paper proposes a new, biologically realistic, computational model for the phenomenon of hippocampal replay. This is an important study with relevance for a broad audience in neuroscience. The proposed model convincingly simulates various aspects of experimental data discovered in the past.

*For correspondence:
sen.cheng@rub.de

**Competing interest:** The authors declare that no competing interests exist.

## Introduction

Humans and other animals continuously make decisions that impact their well-being, be that shortly after emitting a choice or much later. To successfully optimize their behavior, animals must be able to correctly credit choices with resulting consequences, adapt their future behavior, and remember what they have learned. The hippocampus is known to play a critical role in the formation of new memories and retrieval of memories, as evidenced by the famous case of patient H.M. (*Corkin et al., 1997*) and others (*Wilson et al., 1995*; *Rosenbaum et al., 2004*). In rats and mice damage to the hippocampus is known to impair spatial learning and memory (*Morris et al., 1982*; *Deacon et al., 2002*). An important phenomenon linked to learning and memory found in the hippocampus is that of 'replay' (*Buhry et al., 2011*). As an animal navigates in an environment, so-called place cells (*O'Keefe and Dostrovsky, 1971*) in the hippocampus are sequentially activated. Later, during awake resting states and during sleep, compressed reactivation of the aforementioned sequences can be observed within events of high-frequency neural activity which are known as sharp wave/ ripples (*Buzsáki, 1989*; SWRs). These sequences preferentially start at the current position of the animal (*Davidson et al.,*

*2009*) and can occur in the order observed during behavior as well as in the reverse order (*Diba and Buzsáki, 2007*).

Consistent with the proposed function of replay in learning, *Widloski and Foster, 2022* showed that in a goal-directed navigation task replay sequences obey spatial constraints when barriers in an environment change on a daily basis. By contrast, other studies suggest that replay is not limited to just reflecting the animal's previous behavior. For instance, replay can represent shortcuts that the animals had never taken (*Gupta et al., 2010*). Replay during sleep appeared to represent trajectories through reward-containing regions that the animals had seen, but never explored (*Ólafsdóttir et al., 2015*). Following a foraging task, replay sequences resembled random walks, that is, their statistics was described by a Brownian diffusion process, even though the preceding behavioral trajectories did not (*Stella et al., 2019*). How can the hippocampus support the generation of such a variety of replay statistics and how do the sequences facilitate learning? Here, we address these questions by studying replay and its effect on learning using computational modeling.

We adopt the reinforcement learning framework (RL), which formulates the problem of crediting choices with resulting consequences of an agent interacting with its environment and trying to maximize the expected cumulative reward (*Sutton and Barto, 2018*). A common way of solving this problem involves the learning of a so-called value function, which maps pairs of environmental states and actions to expected future rewards, and then choosing actions that yield the highest value. A popular algorithm that can learn such a function is Q-learning, in which the value function is referred to as the Q-function and expected future rewards are referred to as Q-values (*Watkins, 1989*). The Q-function is updated using the so-called temporal-difference (TD) error, which is computed from an experience's immediate reward and the Q-function estimates of the future reward (see Materials and

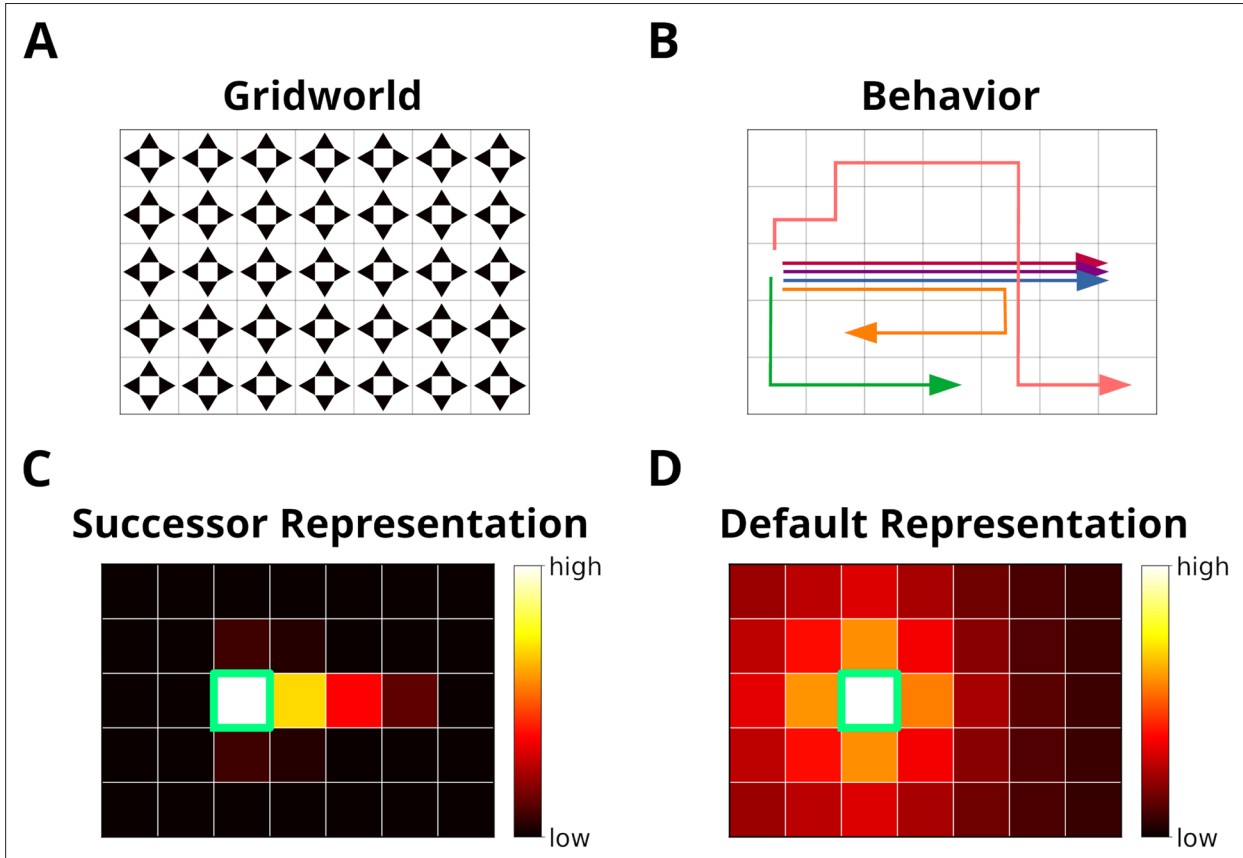

**Figure 1.** The grid world. (**A**) An example of a simple grid world environment. An agent can transition between the different states, that is, squares in the grid, by moving in the four cardinal directions depicted by arrowheads. (**B**) Example trajectories of an agent moving in the grid world. (**C**) The successor representation (SR) for one state (green frame). Note that the SR depends on the agent's actual behavior. (**D**) The default representation (DR) for the same state as in **C**. In contrast to the SR, the DR does not depend on the agent's actual behavior and is equivalent to the SR given random behavior.

methods section for more details). While RL has traditionally been used to solve technical control problems, it has recently been adopted to model animal (*Bathellier et al., 2013*) and human (*Redish et al., 2007*; *Zhang et al., 2018*) behavior.

RL generally requires many interactions with the environment, which results in slow and inefficient learning. Interestingly, replay of stored experiences, that is, interactions with the environment, greatly improves the speed of learning (*Lin, 1992*). The function of experience replay in RL has been linked to that of hippocampal replay in driving spatial learning (*Johnson and Redish, 2005*). *Mattar and Daw, 2018* proposed a model of hippocampal replay as the replay of experiences which have the highest 'utility' from a reinforcement learning perspective, meaning that the experience which yields the highest behavioral improvement is reactivated. In their model, the agent's environment is represented as a so-called grid world, which discretizes space into abstract states between which the agent can move using the four cardinal actions (*Figure 1A*). During experience replay, experiences associated with the different transitions in the environment can be reactivated by the agent. Their model accounts for multiple observed replay phenomena. However, it is unclear how the brains of animals could compute or even approximate the computations required by Mattar and Daw's model. The brain would have to perform hypothetical updates to its network for each experience stored in memory, compute and store the utility for each experience, and then reactivate the experience with the highest utility. Since biological learning rules operate on synapses based on neural activity, it appears unlikely that hypothetical updates can be computed and their outcome stored without altering the network.

Here, we introduce a model of hippocampal replay which is driven by experience and environmental structure, and therefore does not require computing the hypothetical update that a stored experience would lead to, if it were reactivated. Especially during early learning, the sequences generated by our proposed mechanism are often the optimal ones according to Mattar and Daw, and our replay mechanism facilitates learning in a series of spatial navigation tasks that comes close to the performance of Mattar and Daw's model. Furthermore, we show that a variety of hippocampal replay statistics emerges from the variables that drive our model. Hence, our model could be seen as an approximation of Mattar and Daw's model that avoids the computation of hypothetical updates at a small cost to learning performance.

## Results

### Using structural knowledge and the statistics of experience to prioritize replay

We propose a model for the prioritization of experiences, which we call *Spatial structure and Frequency-weighted Memory Access*, or SFMA for short. The model was conceived with simplified grid world environments in mind (*Figure 1A*). Each node in the grid corresponds to an environmental state. During behavior a reinforcement learning agent transitions between nodes and stores these transitions as experience tuples $e_t = (s_t, a_t, r_t, s_{t+1})$. Here, $s_t$ is the current state, $a_t$ is the action executed by the agent, $r_t$ is the reward or punishment received after transitioning, and $s_{t+1}$ is the next state.

Reactivating stored experiences is viewed as analogous to the reactivation of place cells during replay events in the hippocampus. Assuming an experience $e_t$ has just been reactivated, each stored experience $e$ is assigned a priority rating $R(e|e_t)$ based on its strength $C(e)$, its similarity $D(e|e_t)$ to $e_t$ and inhibition $I(e)$ applied to it (*Figure 2A*):

$$R(e|e_t) = C(e)D(e|e_t)[1 - I(e)] \qquad (1)$$

The experience strength $C(e)$ is modulated by the frequency of experience and reward.

The similarity $D(e|e_t)$ between e and $e_t$ reflects the spatial distance between states, taking into account structural elements of the environment, such as walls and other obstacles. This rules out the use of Euclidean distances. A possible candidate would be the successor representation (SR) (*Dayan, 1993*). Given the current state state $s_j$, the SR represents as a vector the likelihood of visiting the other states $s_i$ in the near future (*Figure 1C*). This likelihood is commonly referred to as the discounted future occupancy. Since states separated by a barrier are temporally more distant to each other than to those without a barrier, their expected occupancy is reduced more by discounting. However, the SR suffers from two disadvantages. First, the SR depends on the agent's current behavior, i.e., the

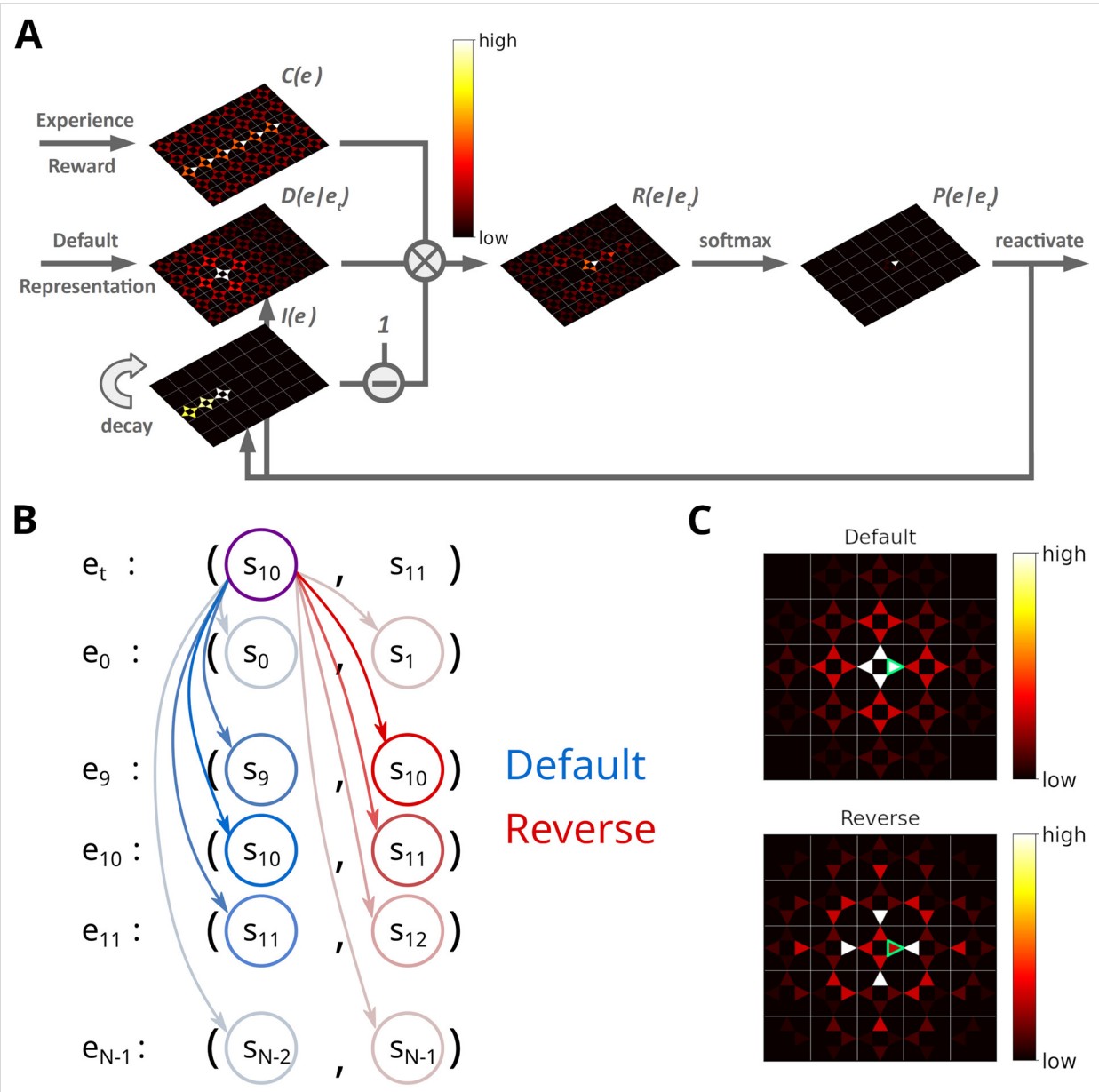

**Figure 2.** Illustration of the Spatial structure and Frequency-weighted Memory Access (SFMA) replay model. (**A**) The interaction between the variables in our replay mechanism. Experience strength $C(e)$, experience similarity $D(e|e_t)$ and inhibition of return $I(e)$ are combined to form reactivation prioritization ratings. Reactivation probabilities are then derived from these ratings. (**B**) Experience tuples contain two states: the current state and next state. In the default mode, the current state of the currently reactivated experience (violet circle) is compared to the current states of all stored experiences (blue arrows) to compute the experience similarity $D(e|e_t)$. In the reverse mode, the current state of the currently reactivated experience is compared to the next state of all stored experiences (red arrows). (**C**) Example of the similarity of experiences to the currently reactivated experience (green arrow) in an open field for the default and reverse modes. Experience similarity is indicated by the colorbar. In the default mode, the most similar experiences are the current state or those nearby. In the reverse mode, the most similar experiences are those that lead to the current state.

The online version of this article includes the following figure supplement(s) for figure 2:

**Figure supplement 1.** Step by step example of a replay sequence generated by SFMA.

**Figure supplement 2.** Reconciling non-local replays and preferential replay for current location in online replay.

**Figure supplement 3.** Possible replay modes and example replay trajectories.

agent's policy, which will distort the distance relationships between states unless that agent behaves randomly. Second, if the structure of the environment changes, the SR has to be relearned completely, and this may be complicated further by the first problem. These problems were recently addressed by *Piray and Daw, 2021* in the form of the default representation (DR) (*Figure 1D*). Unlike the SR, the DR does not depend on the agent's policy, but rather on a default policy, which is a uniform action policy, that is, random behavior, and the DR was shown to reflect the distance relationships between states even in the presence of barriers. Importantly, *Piray and Daw, 2021* demonstrated that the DR can be efficiently updated if the environmental structure changes using a low-rank update matrix. We chose to base experience similarity on the DR due to its advantageous features.

Since experience tuples contain two states – the current state $s_t$ and the next state $s_{t+1}$ – we consider two ways of measuring the similarity between experiences. In the default mode, we compare the current states of two experiences (*Figure 2B*). In the reverse mode, the current state of the most recently reactivated experience is compared to the next states of the other experiences. We found that default and reverse modes tend to generate different kinds of sequences. Two more replay modes could be defined in our model, which we will not consider in this study. We explain this choice in the Discussion.

After an experience $e$ has been reactivated, inhibition $I(e)$ is applied to prevent the repeated reactivation of the same experience. Inhibition is applied to all experiences sharing the same starting state $s_t$, i.e., inhibition is set to $I(e) = 1$, and decays in each time step by a factor of $0 < \lambda < 1$. Inhibition values are maintained for one replay epoch and are reset when a new replay epoch is initiated.

The next experience to be reactivated is randomly chosen according to the activation probabilities $P(e|e_t)$, which are computed by applying a customized softmax function on $R(e|e_t)$ (see Materials and methods). If the priority ratings for all experiences falls below a threshold $\theta = 10^{-6}$ replay is stopped. The definition of one replay epoch is summarized in Algorithm 1. Sequences of replay are produced by iteratively activating and inhibiting individual experiences. Together, experience strengths and experience similarities guide replay while inhibition promotes the propagation of sequences (*Figure 2— figure supplement 1*).

To initiate replay, we consider two cases: replay during awake resting states (online replay) is initiated at the agent's current position, whereas replay during sleep (offline replay) is initiated with an experience randomly selected from memory based on the relative experience strengths. We chose different initialization schemes since awake replay has been reported to start predominantly at the animal's current position (*Davidson et al., 2009*). However, there are also non-local sequences in awake replay (*Karlsson and Frank, 2009*; *Gupta et al., 2010*; *Ólafsdóttir et al., 2017*). Non-local replays are generated by our model in offline replay, albeit with a weaker bias for the current position (*Figure 2—figure supplement 2A, B*). To model awake replay, we could increase the bias in the random initialization by raising the experience strength for experiences associated with the current position (*Figure 2—figure supplement 2C–F*), while preserving some non-local replays. However, for simplicity we opted to simply initiate awake replay at the current location.

---

**Algorithm 1. Spatial structure and Frequency-weighted Memory Access (SFMA)**

---

Require:  $e_t$ (Replay initiated)
1: **for** t=1:N **do**
2:     **for** experience $e$ **do**
3:         Compute priority rating $R(e|e_t) = C(e)D(e|e_t)[1 - I(e)]$.
4:     **end for**
5:     **if** $\max R < \theta$ **then**
6:         Stop replay
7:     **end if**
:      Compute reactivation probabilities $P(e|e_t)$.
9:     Choose next experience $e_{t+1}$ to reactivate.
10:    Reduce/decay inhibition for all stored experiences.
11:    Inhibit experience $e_{t+1}$.
12: **end for**

---

## SFMA facilitates spatial learning

We begin by asking what benefit a replay mechanism such as implemented in SFMA might have for spatial learning. To do so, we set up three goal-directed navigation tasks of increasing difficulty: a linear track, a square open field, and a labyrinth maze (*Figure 3A*). Simulations were run for the default

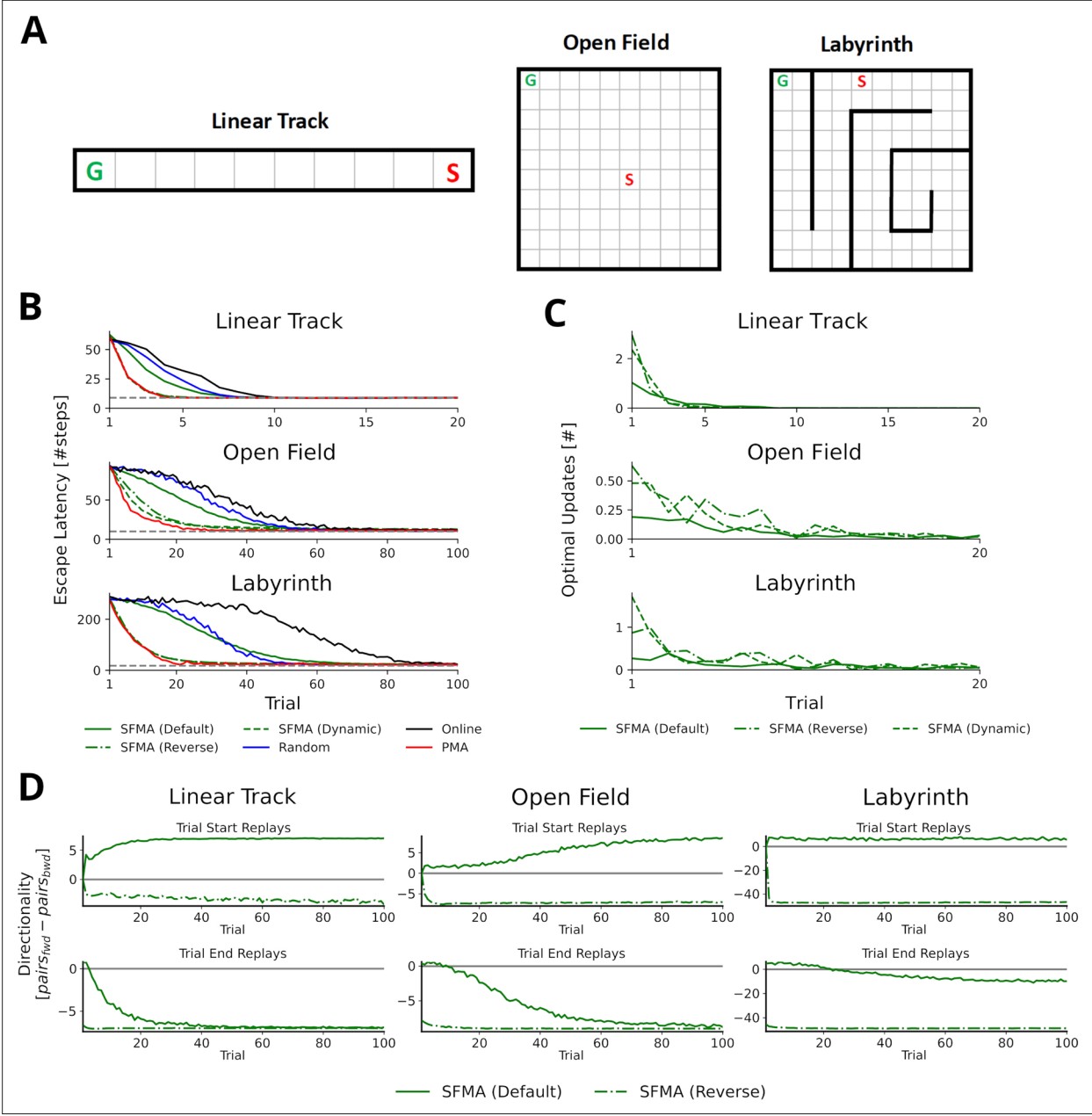

**Figure 3.** Statistics of replay has large impact on spatial learning. (**A**) The three goal-directed navigation tasks that were used to test the effect of replay on learning: linear track, open field and maze. In each trial, the agent starts at a fixed starting location S and has to reach the fixed goal location G. (**B**) Performance for different agents measured as the escape latency over trials. Shown is the performance for an online agent without replay (black), an agent that was trained with random replay (blue), our SFMA model (green), and the Prioritized Memory Access model (PMA) by *Mattar and Daw, 2018* (red). The results of our SFMA model are further subdivided by the replay modes: default (solid), reverse (dash-dotted), and dynamic (dashed). Where the dashed and dash-dotted green lines are not visible they are overlapped by the red solid line. (**C**) Reverse and dynamic replay modes produce more optimal replays while the default replay mode yields pessimistic replays. Shown is the number of optimal updates in the replays generated on each trial for different replay modes: default (solid), reverse (dash-dotted), and dynamic (dashed). Note, that in later trials there is a lack of optimal updates because the learned policy is close to the optimal one and any further updates have little utility. (**D**) Directionality of replay produced by the default (solid) and reverse (dash-dotted) modes in the three environments. The reverse replay mode produces replays with strong reverse directionality irrespective of when replay was initiated. In contrast, the default mode produces replays with a small preference for forward directionality. After sufficient experience with the environment the directionality of replays is predominantly forward for replays initiated at the start and predominantly reverse for replays initiated at the end of a trial.

The online version of this article includes the following figure supplement(s) for figure 3:

**Figure supplement 1.** Directionality of consecutive replay pairs.

*Figure 3 continued on next page*

**Figure supplement 2.** Example replay trajectories in different spatial navigation tasks.

**Figure supplement 3.** Reward changes trigger higher probability of activating the reverse mode.

**Figure supplement 4.** The number of sequences decreases with experience.

and reverse replay modes and compared to an agent trained without experience replay, an agent trained with random experience replay, and the state-of-the-art Prioritized Memory Access model (PMA) by *Mattar and Daw, 2018*.

Using the reverse mode, our model clearly outperformed the agents trained without replay and with random replay (*Figure 3B*). Importantly, performance was close to, even if slightly below, that of PMA. Learning performance of SFMA critically depends on the replay mode. The default mode yielded a performance that was much lower than the reverse mode and only slightly better than random replay. Considering its low performance, the default mode may appear undesirable, however, it is important for two reasons. First, it generates different types of sequences from the reverse mode (*Figure 3D* and *Figure 3—figure supplement 1*) and these sequences are more consistent with some experimental observations. While the reverse mode mostly generates reverse sequences in all situations, the default mode generates forward replay sequences at the beginning of trials and reverse sequences at the end of trials. This pattern of changing replay directions has been observed in experiments in familiar environments (*Diba and Buzsáki, 2007*). Also, as we will show below the default mode generates shortcut replays like those found by *Gupta et al., 2010*, whereas the reverse mode does not. Second, the recent discovery of so-called pessimistic replay, that is, replay of experiences which are not optimal from a reinforcement learning perspective, show that suboptimal replay sequences occur in human brains (*Eldar et al., 2020*) for a good reason (*Antonov et al., 2022*). Such suboptimal replays are better supported by the default mode (*Figure 3C*).

We therefore suggest that the reverse and default modes play distinct roles during the learning progress. Early in the learning session or following reward changes, when the environment is novel and TD errors are high, the reverse mode is preferentially used to acquire a successful behavior quickly (*Figure 3—figure supplement 3*). Indeed, in our simulations the reverse mode produces more updates that were optimal, that is, they improved the agent's policy the most, than did the default mode (*Figure 3C*). The preponderance of reverse replay sequences during early learning is consistent with experimental observations in novel environments (*Foster and Wilson, 2006*) or after reward changes (*Ambrose et al., 2016*). Later in the learning session, when the environment has become familiar and TD errors are low, the default mode is preferentially used to make the learned behavior more robust. The default mode then accounts for interspersed reverse and forward replay sequences in familiar environments. We provide a more detailed rationale for the default mode in the Discussion. We call this strategy of switching between the reverse and default modes the *dynamic* mode of SFMA. Put simply, in the dynamic mode the probability of generating replay using the reverse mode increases with the TD errors accumulated since the last trial (for more details see Materials and methods). It yields a learning performance (*Figure 3B*) and number of optimal updates (*Figure 3C*) that are similar to the reverse mode.

In the following, we focus on the statistics of replay that SFMA generates in a range of different experimental settings.

## Near-homogeneous exploration of an open environment explains random walk replays

We first investigated the replay statistics that our model produces in a simple environment without navigational goals similar to the experiment of *Stella et al., 2019*. To this end, we created a virtual square grid world environment of size 100 × 100 without rewards. For simplicity, we first set experience strengths to the same value, $C(e) = 1$, to reflect homogeneous exploration of the environment. All replays were generated using the default mode since the environment was familiar to the animal in the experiments, but using the reverse mode did not affect the results (*Figure 4—figure supplement 1*).

With this setup the trajectories represented by the replayed experiences of the agent are visually similar to random walks (*Figure 4A*). The displacement distribution of replayed locations indicates

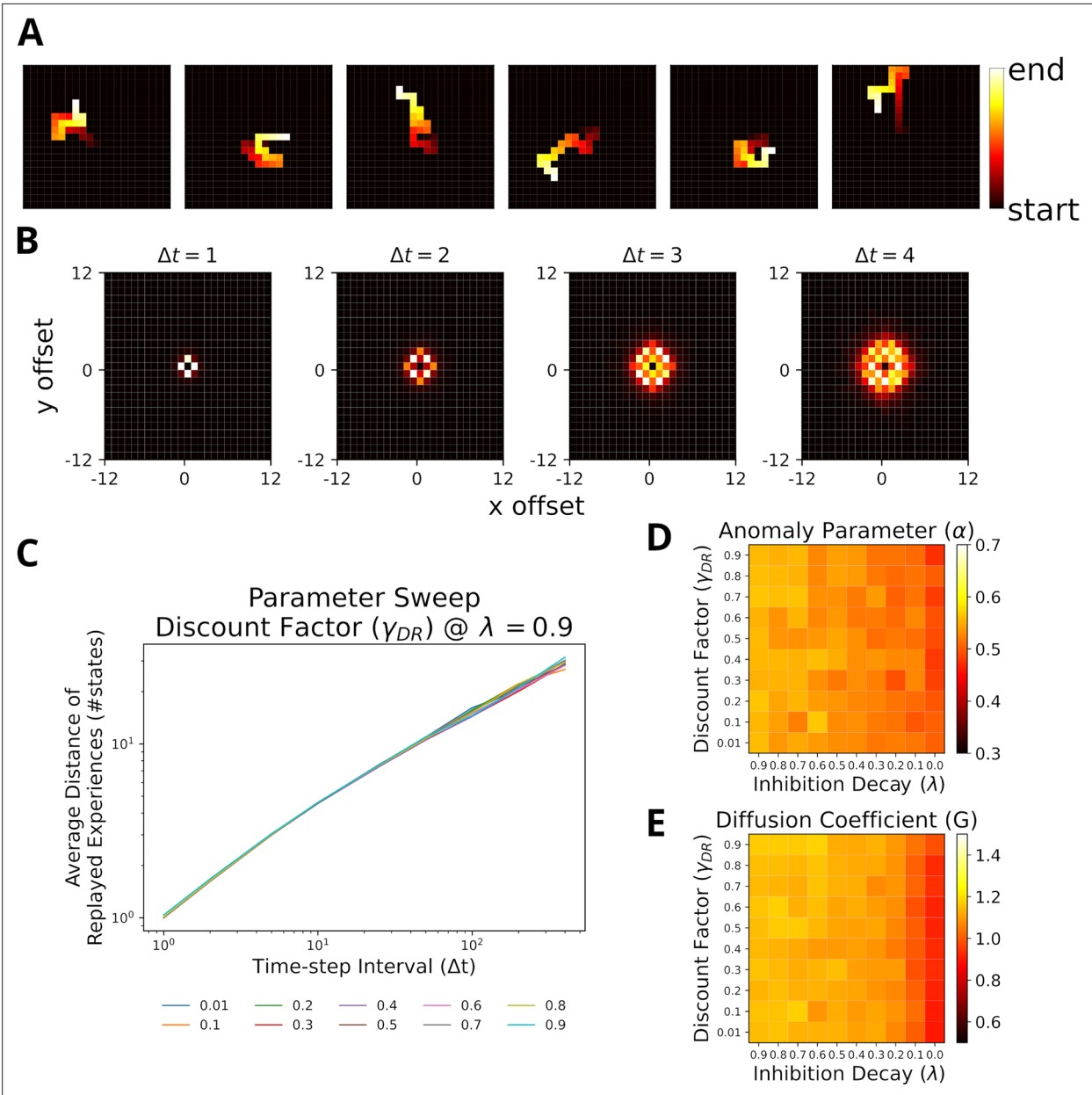

**Figure 4.** Replays resemble random walks across different parameter values for Default Representation (DR) discount factor and inhibition decay. (**A**) Example replay sequences produced by our model. Reactivated locations are colored according to recency. (**B**) Displacement distributions for four time steps (generated with $\beta_M = 5$). (**C**) A linear relationship in the log-log plot between average distance of replayed experiences and time-step interval indicates a power-law. Lines correspond to different values of the DR's discount factor $\gamma_{DR}$ as indicated by the legend. (**D**) The anomaly parameters (exponent $\alpha$ of power law) for different parameter values of DR and inhibition decay. Faster decay of inhibition, which allows replay to return to the same location more quickly, yields anomaly parameters that more closely resemble a Brownian diffusion process, that is, closer to 0.5. (**E**) The diffusion coefficients for different parameter values of DR and inhibition decay. Slower decay of inhibition yields higher diffusion coefficients.

The online version of this article includes the following figure supplement(s) for figure 4:

**Figure supplement 1.** Replays generated using the reverse mode also resemble random walks across different parameter values.

**Figure supplement 2.** Displacement distributions for different inverse temperature values.

**Figure supplement 3.** For heterogeneous experience strengths replays also resemble random walks across different parameter values.

**Figure supplement 4.** The starting positions of replay are randomly distributed across the environment.

**Figure supplement 5.** Prioritized Memory Access' (PMA) ability to produce sequences is severely disrupted when the gain calculation for n-step updates is adjusted.

that replay slowly diffuses from its starting location (*Figure 4B* and *Figure 4—figure supplement 2*). To analyze the trajectories more systematically, we used the Brownian Diffusion Analysis also used by *Stella et al., 2019*. In this analysis, a random walk is described by a power law relationship between the average distance between two replayed positions $\Delta x$ and the time interval $\Delta t$, i.e., $\Delta x = G\Delta t^{\alpha}$ with $\alpha = 0.5$. Indeed, the simulated replays exhibit a linear relationship in the log-log-plot indicating a power law between the two variables (*Figure 4C*) and the slope is close to the theoretical value for a random walk $\alpha = 0.5$. This result is robust across a large range of model parameters, the most relevant are the DR's discount factor $\gamma_{DR}$ and inhibition decay $\lambda$, and the range of values in our simulations, $\alpha \in [0.467, 0.574]$ (*Figure 4D*), is a good match to the values reported by *Stella et al., 2019*. ($\alpha \in [0.45, 0.53]$). The values for the diffusion coefficient (*Figure 4E*), which relate to the reactivation speed, are similarly robust and only affected when the decay factor is close to zero. Hence, our model robustly reproduces the experimental findings.

Since the results above were obtained assuming homogeneous exploration of the environment, we repeated our simulations with heterogeneous experience strengths (*Figure 4—figure supplement 3B*). While the relationships in the log-log-plot seems to slightly deviate from linear for very large time-step intervals, the statistics still largely resemble a random walk (*Figure 4—figure supplement 3C*). The range of values for $\alpha \in [0.410, 0.539]$ still covers the experimental observations, albeit shifted toward smaller values.

Stella et al. further reported that the starting locations of replay were randomly distributed across the environment and replay exhibited no preferred direction. Our model reproduces similar results in the case of homogeneous experience strengths (*Figure 4—figure supplement 4A, B*) and heterogeneous experience strengths (*Figure 4—figure supplement 4C, D*). The results of our simulations suggest that replay resembling random walks can be accounted for given near-homogeneous exploration of an open-field environment. If exploration is non-homogeneous, the statistics of a random walk hold only for short to medium time-step intervals.

## Stochasticity results in shortcut replays following stereotypical behavior

*Gupta et al., 2010* provided further evidence that replay does not simply reactivate previously experienced sequences, by showing that replay sequences sometimes represent trajectories that animals were prevented from taking. These so-called shortcut sequences were synthesized from previously experienced trajectories. We constructed a simplified virtual version of the Gupta et al. experiment (*Figure 5A*) to test whether, and under which conditions, our proposed mechanism can produce shortcut replays. In the experiment, animals exhibited very stereotypical behavior, that is, they ran laps in one direction and were prevented from running back. Therefore, in our model the agent was forced to run one of three predefined patterns: right laps, alternating laps, and left laps (*Figure 5B*). This allowed us to focus on the effect of different specific behavioral statistics on replay. The virtual agent was made to run 20 trials in one session and replays were simulated after each trial.

In the first 10 trials, the agents ran in one pattern, and in the next 10 trials, the agents used the same or a different pattern. To model Gupta et al.'s study, we ran simulations with the combinations right-left, right-alternating, and alternating-left. In addition, the combination alternating-alternating served as a baseline condition in which the left and right laps had roughly the same amount of experience throughout the simulation.

In this case, only the default mode could reproduce the experimental findings, while the reverse mode could not. The main cause lay within the large dissimilarity between the experiences at the decision point (D in *Figure 5A*) and the experiences to the left and right of it, which in turn drove prioritization values to zero.

We found that our model produces shortcut-like replays for each run pattern combination (*Figure 5C and D*). Shortcuts occurred in higher numbers for right-left and right-alternating combinations, and mainly in the last 10 trials, in trials when the agent was running a left lap, that is, every trial for right-left and every other trial for right-alternating (*Figure 5—figure supplement 1*). Across the whole running session, for alternating-alternating and alternating-left combinations a lower number of shortcuts occurred. This difference between combinations results from the balance of experience strengths associated with either of the laps and with the center piece. For right-alternating and right-left combinations the experience strengths associated with right lap and center piece were similar

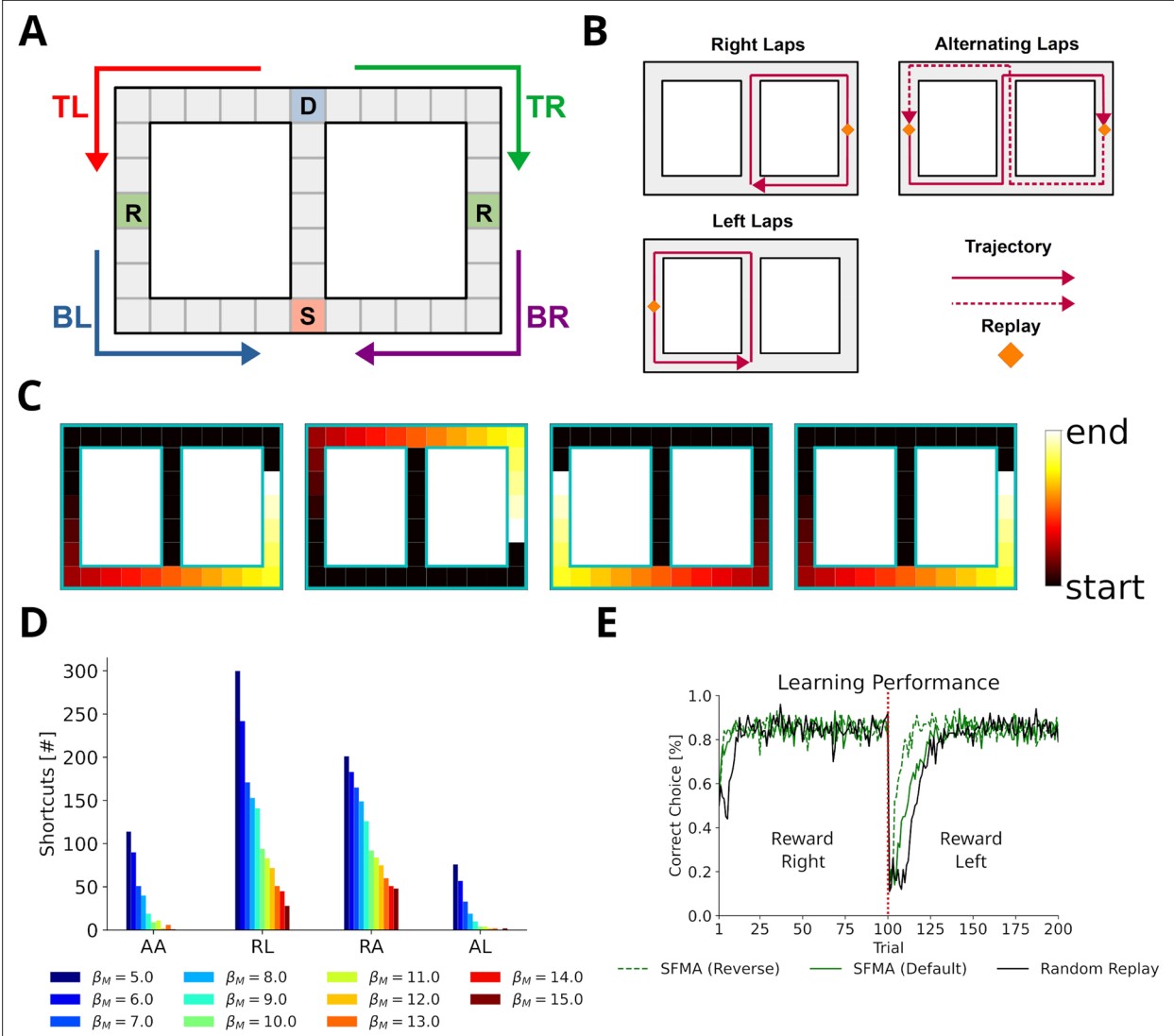

**Figure 5.** Replay of shortcuts results from stochastic selection of experiences and the difference in relative experience strengths. (**A**) Simplified virtual version of the maze used by *Gupta et al., 2010*. The agent was provided with reward at specific locations on each lap (marked with an R). Trials started at bottom of the center corridor (marked with an S). At the decision point (marked with a D), the agent had to choose to turn left or right in our simulations. (**B**) Running patterns for the agent used to model different experimental conditions: left laps, right laps and alternating laps. Replays were recorded at the reward locations. (**C**) Examples of shortcut replays produced by our model. Reactivated locations are colored according to recency. (**D**) The number of shortcut replays pooled over trials for different running conditions and values of the inverse temperature $\beta_M$. Conditions: alternating-alternating (AA), right-left (RL), right-alternating (RA), and alternating-left (AL). (**E**) Learning performance for different replay modes compared to random replay. The agent's choice depended on the Q-values at the decision point. The agent was rewarded for turning right during the first 100 trials after which reward shifted to the left (red line marks the shift).

The online version of this article includes the following figure supplement(s) for figure 5:

**Figure supplement 1.** Number of shortcut replays on a trial by trial basis differs depending on behavioral statistics.

**Figure supplement 2.** Experience strengths resulting from different behavioral statistics in *Gupta et al., 2010* experiment.

**Figure supplement 3.** Following strongly stereotypical behavior efficient learning occurs for the default mode after a change in goal location.

and therefore replay, when initiated at the left lap's reward location, was not biased to reactivate experiences along the center piece. The number of shortcut replays then decreased as the experience strengths associated with the center piece increased relative to those associated with either of the laps. Since for alternating-left and alternating-alternating combinations the experience strengths associated with the center piece already were higher relative to those associated with the laps an overall low but constant number of shortcut replays was produced.

Across all conditions the number of shortcut replays decreased with increasing inverse temperature $\beta_M$, which transforms priority ratings to reactivation probabilities. The larger $\beta_M$, the more deterministic replay becomes. This result therefore suggests that shortcut replay is driven by stochasticity in the prioritization process (*Figure 5D* and *Figure 5—figure supplement 1*) and differences in relative experience strengths (*Figure 5—figure supplement 2*) of the maze's center piece and arms. These differences are in turn induced by the specific behavioral statistics.

Finally, we wanted to know if replay facilitates learning in this task. Since the agent cannot remember, which lap it ran in a previous trial, it would not be able to learn with alternating rewarded locations. Therefore, we chose the right-left condition as the learning task. To deal with problematic behavior of the simple RL agents mentioned above we adjusted our simulation to only consider the action selected at the decision point and let the agent run straight otherwise. We ran simulations in default and reverse mode. An agent trained with random experience replay was used as the baseline. All agents solved the task and successfully learned the new goal location after the rewarded locations changed from right to left (*Figure 5E*). Both default and reverse mode yielded faster learning than random experience replay. However, there was no large difference between the replay modes during the first half. These small differences in learning performance suggest that replaying strictly in reverse mode may not be necessary in tasks with stereotypical behavior since the resulting experience strengths would naturally guide replays in forward or reverse depending on where replay started (*Figure 5—figure supplement 3*).

## Dynamic structural modulation allows replay to adapt to environmental changes

Recently, *Widloski and Foster, 2022* reported that hippocampal replay dynamically adapts to daily changes in the structure of the environment. We argue that this type of replay can be accounted for by representing the environmental structure dynamically with the default representation (DR) (*Piray and Daw, 2021*). While the SR (*Dayan, 1993*) could also be used to represent the environment's structure, it cannot be updated efficiently in the face of structural changes and depends on the agent's

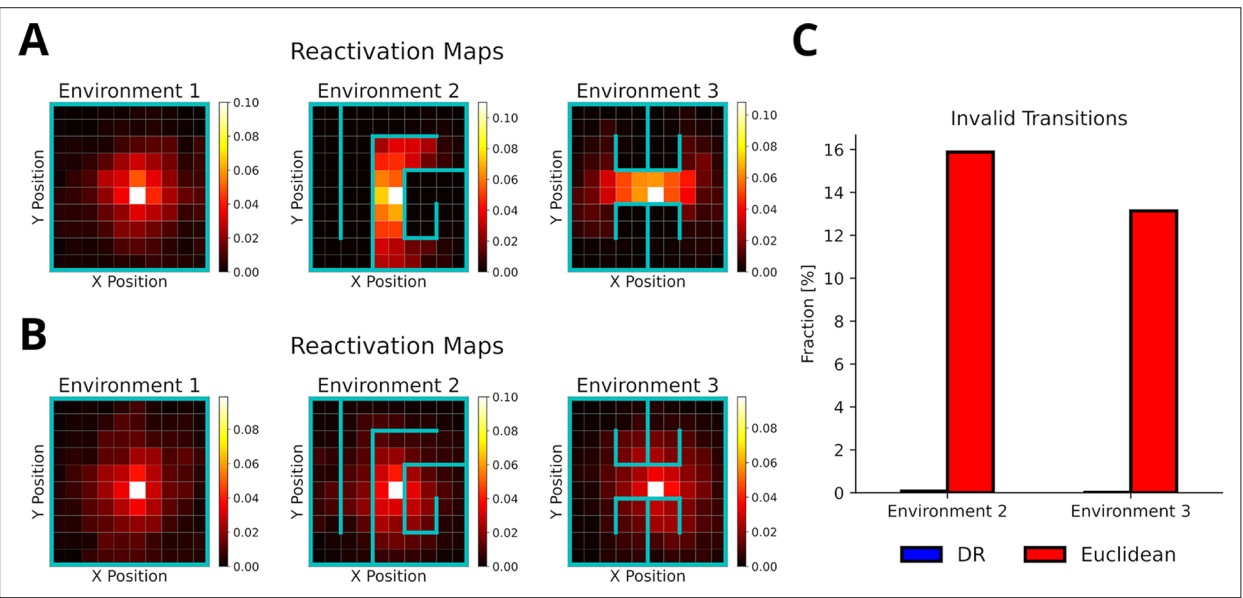

**Figure 6.** The default representation (DR) allows replay to adapt to environmental changes. (**A**) Reactivation maps, that is, the fraction of state reactivations, for all replays recorded in three environments while the agent is located at the white square. Experience similarity is based on the DR with a discount factor $\gamma_{DR} = 0.1$. The colorbar indicates the fraction of reactivations. Note that replay obeys the current environmental boundaries. (**B**) Same as in A, but experience similarity is based on the Euclidean distance between states. Note that replay ignores the boundaries. (**C**) The fraction of invalid transitions during replay in different environments for the DR (blue) and Euclidean distance (red). While replay can adapt to environmental changes when using the DR, it does not when experience similarity is based on the Euclidean distance.

The online version of this article includes the following figure supplement(s) for figure 6:

**Figure supplement 1.** Reverse mode replays adapt to environmental changes.

policy. The DR does not suffer from these problems. Specifically, we employed the DR by separating the spatial representation into open 2D space and information about environmental barriers. Barrier information was provided when the environment changed (see Methods).

To test whether our proposed replay mechanism can adapt to changes of the environment's structure, we simulated replays in three distinct consecutive environments (*Figure 6A*) and evaluated the occurrence of replay crossing environmental barriers. All replays were generated using the default mode, but using the reverse mode did not affect these results (*Figure 6—figure supplement 1*). Our proposed replay mechanism successfully adapts to changes in the environment's structure as illustrated by replays diffusing along barriers in the reactivation maps (*Figure 6A*) and shown by the absence of invalid transitions (*Figure 6C*). In contrast, when experience similarity is based on the Euclidean distance between locations, replay does not conform or adapt to the changing environmental structure (*Figure 6B and C*).

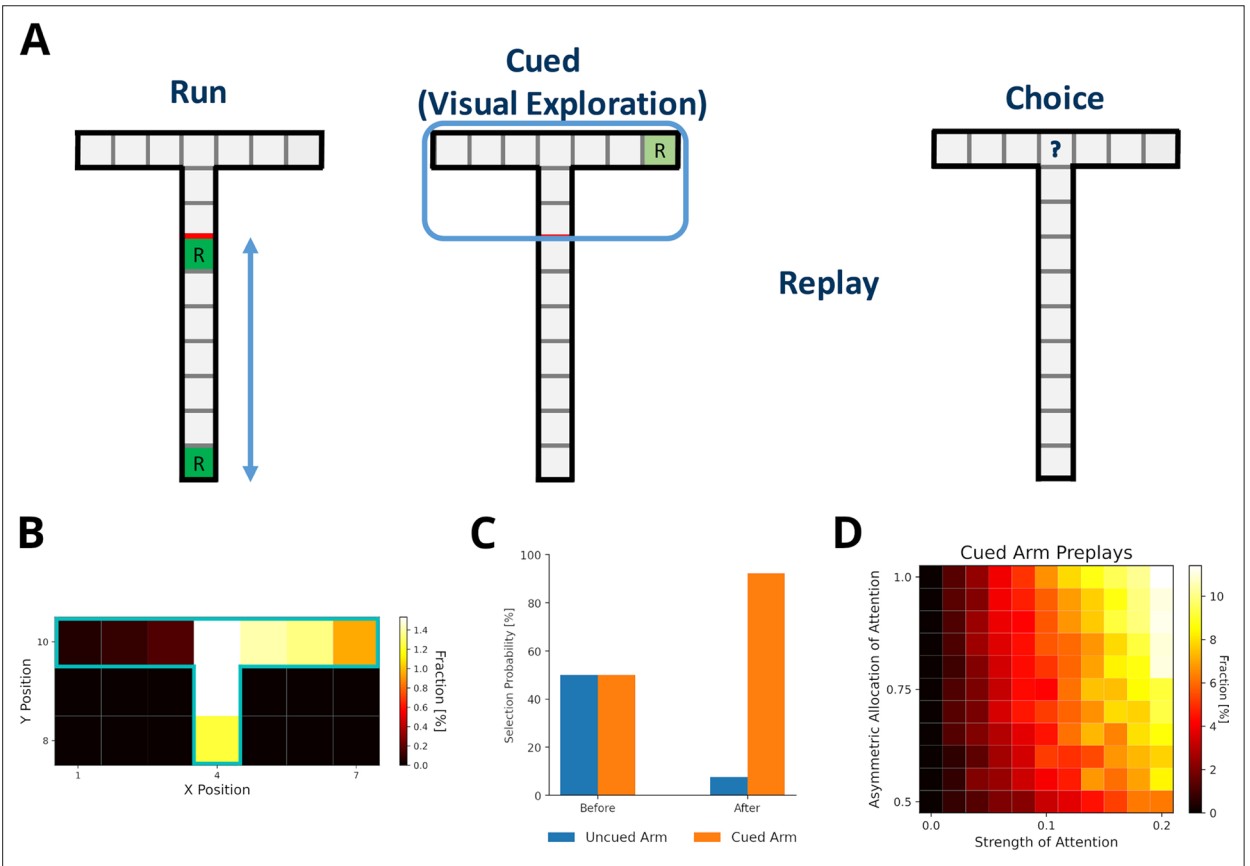

**Figure 7.** Preplay of cued, but unvisited, locations can be explained by visual exploration. (**A**) Schematic of the simplified virtual maze and task design used to model study by *Ólafsdóttir et al., 2015*. First, in an initial run phase, the agent ran up and down the stem of the maze. Second, in the following visual exploration phase, the agent visually explores a cued part of the maze and increases the strength of the associated experiences. Third, experiences are reactivated according to their priority scores. Fourth, agents are given access to the entire T-maze. The fraction of choosing the two arms in the test phase are compared. (**B**) Fractions of reactivated arm locations. Experiences associated with the cued arm (right) are preferentially reactivated. (**C**) The fractions of choosing either arm before and after training the agent with preplay. Before preplay the agent shows no preference for one arm over the other. After preplay, the agent preferentially selects the cued arm over the uncued arm. (**D**) The percentage of cued-arm preplay (out of all arm preplays and stem replays) for different amounts of attention paid to the cued arm vs. the uncued arm (asymmetric allocation of attention) and experience strength increase relative to actual exploration (attention strength).

The online version of this article includes the following figure supplement(s) for figure 7:

**Figure supplement 1.** Preplay of yet unvisited cued locations can be explained by visual exploration (reverse mode).

## Preplay of unvisited locations requires structural modulation and visual exploration

*Ólafsdóttir et al., 2015* reported the preplay of trajectories through a part of a T-maze that the animal could see, but not physically explore, and that contained a visible reward. Our model can provide an intuitive explanation for the reported phenomenon. First, the use of DR explains, in principle, why the preplay of unvisited locations is possible. The DR allows for the representation of the environment's structure provided it is observable. Still, these experiences would have zero experience strength in our model and therefore would not be reactivated. However, during the reward cueing phase, the animals were reported to have paid attention to the reward cue. We view this as a form of visual exploration that increases the strength of experiences along the attended arm and updates the reward information related to the experiences.

We set up a virtual version of the *Ólafsdóttir et al., 2017* task (*Figure 7A*) in which the agent first runs up and down the maze stem ten times with rewards provided at each end of the stem (run phase). In the cued phase, we increased experience strengths of the blocked arms by a small amount relative to actual exploration (attention strength), which was larger for the cued than the uncued arm (asymmetric allocation of attention). Then experiences were preplayed. Template matching was used to detect the preplay of cued and uncued arms, and replay of the stem. All preplays were generated using the default mode, but results were similar when using the reverse mode (*Figure 7—figure supplement 1*). We found that our model indeed produces preplays of the cued arm and only rarely of the uncued arm (*Figure 7B*). As expected, the attention strength as well as the asymmetric allocation of attention are key to produce preplays (*Figure 7D*). The higher the value of these two parameters, the more often the cued arm is preplayed.

Furthermore, *Ólafsdóttir et al., 2017* reported that animals preferred the cued arm when the barrier blocking the arms was removed and they were given physical access to the maze. Our model shows the same preference for the cued arm after training an agent with simulated replays and testing it in the T-maze (*Figure 7C*).

## Reward modulation of experience strength leads to over-representation of rewarded locations

Replay is known to over-represent rewarded locations (*Singer and Frank, 2009*; *Pfeiffer and Foster, 2013*). This might be accounted for in our model, since the experience strength $C(e)$ is modulated not only by the frequency of experience but also by the frequency of reward. However, it is not intuitively clear how their relative contribution affect the statistics of replay. We therefore tested different values for reward modulation, that is, the increase of experience strength due to reward, in goal-directed navigation tasks in open field and T-maze environments. The default value used in other simulations was one. These simuluations were compared to those with a reward modulation of ten. Note, in our model changing the reward modulation is equivalent to changing the magnitude of reward.

Training lasted for 100 trials. Online replays were generated at the beginning and end of each trial. Additionally, offline replays were also generated after each trial. However, only online replays generated at the end of a trial were used to train the agent. All replays were generated using the default mode.

When the frequency of experience and reward modulate experience strength equally, replays mainly represent locations along the path leading to the rewarded location (*Figure 8A and B*). Replays in the open-field environment over-represent the starting location. The rewarded location shows stronger over-representation for online replays occurring at the end of trials partly due to initialization bias at the goal location. In the T-maze, these effects are less pronounced due to replay being constrained by the maze's structure. Increasing the reward modulation to ten, drives replays to over-represent the rewarded location (*Figure 8C and D*). For online replays at the beginning of trials, this effect is not apparent due to the initialization bias at the starting location. For online replays at the end of trials and offline replay, the representation of the path leading to the rewarded location is heavily reduced and that of the goal location is enhanced.

To summarize, SFMA can account for the over-representation of rewarded locations reported in the literature and this over-representation depends on the balance of modulation due to experience and reward.

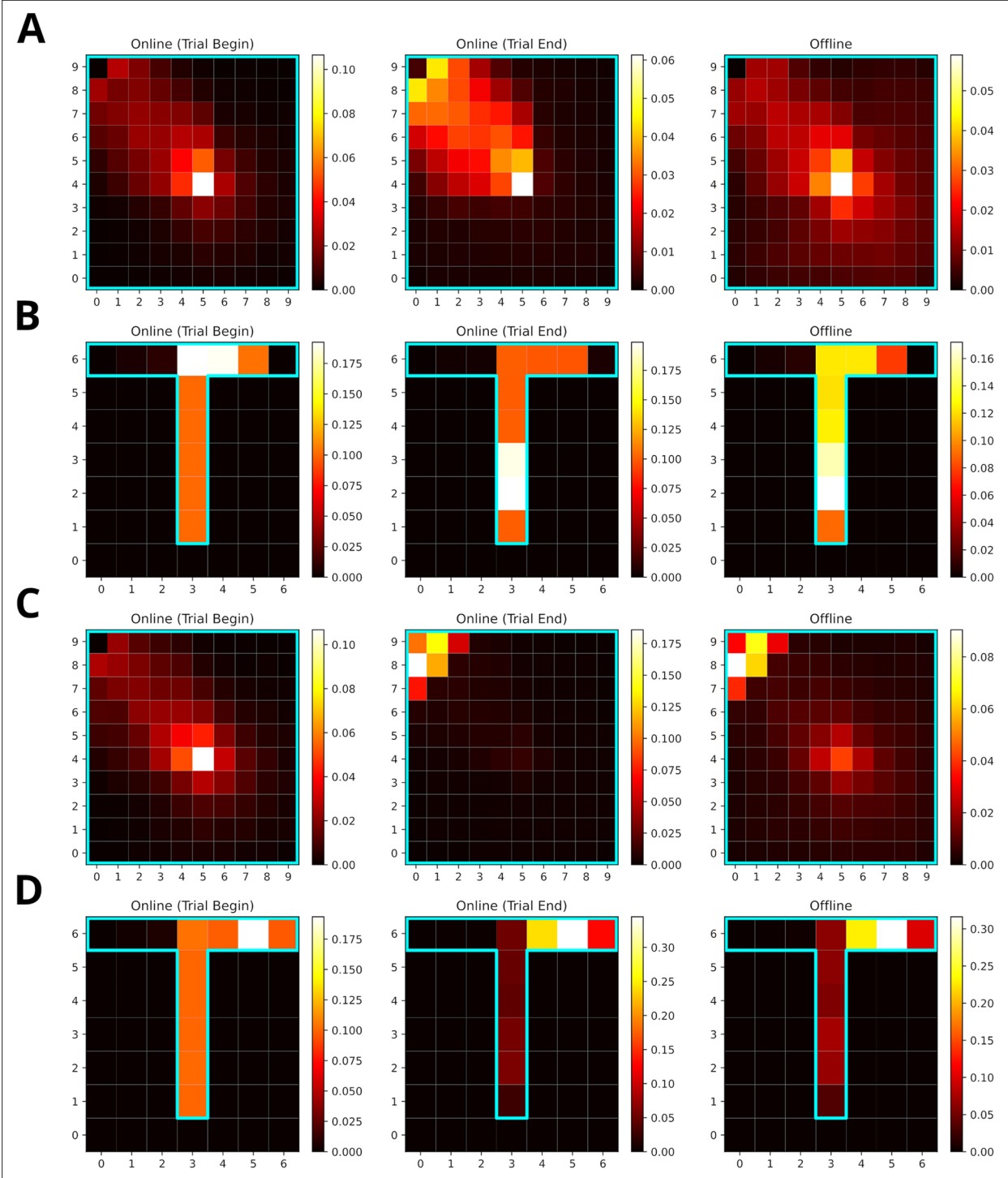

**Figure 8.** Increasing the reward modulation of experience strengths leads to an over-representation of rewarded locations in replays. (**A**) Reactivation maps for online replays generated in an open field environment at trial begin (left) and end (middle) as well as for offline replays (right). Reward modulation was one. Replay tends to over-represent the path leading to the reward location. The starting location is strongly over-represented for all replay conditions, while the the reward location is mainly over-represented for trial end replays. (**B**) Same as A, but in a T-maze. (**C**) Same as A, but due to a higher reward modulation of ten the rewarded location is strongly over-represented in replay. (**D**) Same as C, but in a T-maze.

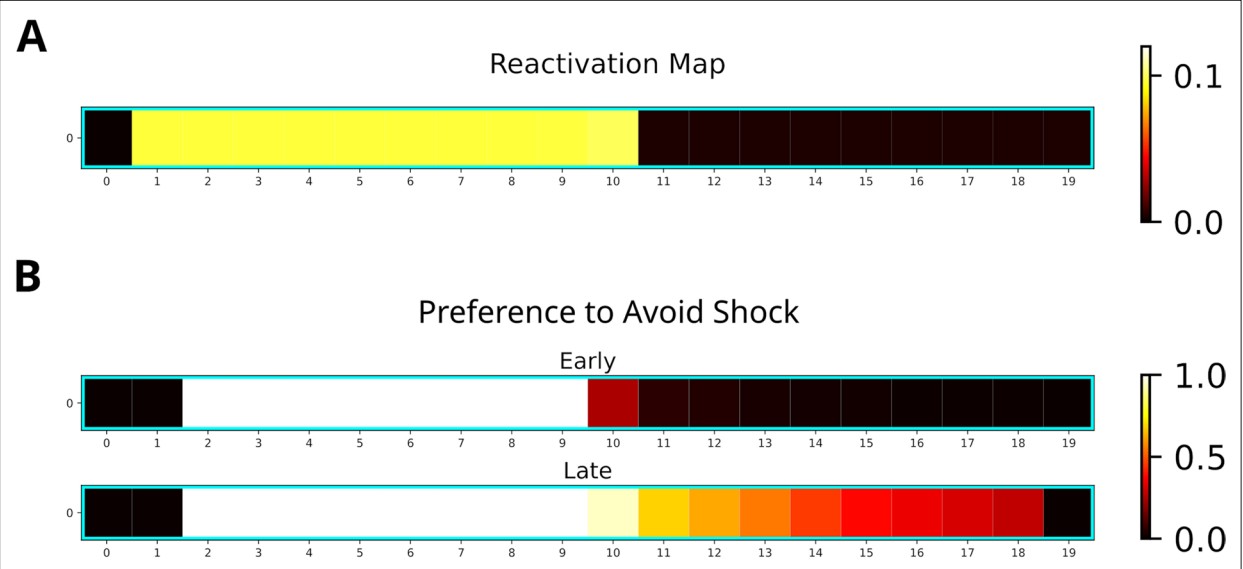

**Figure 9.** Replay will enter aversive zones given previous experience. (**A**) The reactivation map indicates that the dark zone (left half), which contains the shock zone, was preferentially replayed even after the shock administered and the agent stopped in the middle. (**B**) Probability of the Q-function pointing away from the shock zone for each state. After the first five trials (Early), avoidance is apparent only for states in the dark zone (excluding the shock zone itself). In the following five trials (Late), the preference for avoidance is propagated to the light zone (right half).

### Replay enters aversive locations given previous experience, driving the learning of avoidance

Replay was found to not only form reward-directed sequences, but also, rather counter-intuitively, to enter locations associated with aversive experiences (*Wu et al., 2017*). The specific location and amount of replay reported by *Wu et al., 2017* is not within the scope of our model. Nonetheless, SFMA can give a simple account for why replay would enter aversive locations. In their study, the animals had substantial experience with the whole environment, which consisted of a dark and light zone, over three phases (Run 1, Run 2, and Pre) before shock was introduced in the dark zone. Importantly, the animals initially displayed a strong bias for the dark zone.

For our simulation, we used a 20-states-long linear track where the left and right half represented dark and light zones, respectively. The left most two states represented the shock zone. We modeled the initial three phases by having an agent run 30 laps on the linear track environment and multiplied the experience strengths associated with the dark and light zones with the occupancy reported by *Wu et al., 2017*. The remainder of the simulation consisted of a single shock trial in which the agent entered the shock zone and received shock of magnitude one, and nine trials in which the agent ran laps in the light zone. Twenty replays of length ten were generated after receiving the shock. While running laps in the light zone, replays were generated when the agent stopped in front of the dark zone. The generated replays preferentially entered the dark zone and reached the shock zone (*Figure 9A*). This is due to the agent's previous experience in the dark zone, which was over-represented relative to the light zone.

Additionally, we investigated the effect of the generated replays on behavior by training the agent and analyzing the policy. The learned policy preferentially points away from the shock zone (*Figure 9B*), indicating that the agent learned to avoid the shock zone. Avoidance is stronger following more replay and spreads to the light zone (*Figure 9B*, bottom map). Our simulations suggest that previous experience is sufficient for replay to produce sequences that extend to aversive locations and that these sequences could further facilitate the learning of avoidance behavior.

## Discussion

We introduced a replay mechanism (SFMA) that accounts for a variety of different replay statistics reported in the literature. The core aspect of SFMA is the prioritization of experiences according to

the frequency at which they occur, the likelihood with which they are rewarded, and the structure of the environment. Furthermore, reactivated experiences are inhibited in the prioritization process to drive the generation of sequential activations. The statistics of these sequences depend on whether SFMA operates in the default or reverse mode. Surprisingly, despite these differences both modes are consistent with diverse experimental results – with the notable exception of shortcut replays, which arise only in the default mode. However, the two modes have very different impact on spatial learning, which is much faster when using the reverse mode. In the following we discuss some details of our modeling assumptions and results.

## The components of the spatial structure and frequency-weighted memory access (SFMA) model

The prioritization of experiences during replay is well supported by experimental evidence. Behavioral data suggests that animals use spatial representations to quickly adapt to changes in the environment (*de Cothi et al., 2020*). Structural modulation of hippocampal replay is reported in the literature (*Wu and Foster, 2014*; *Ólafsdóttir et al., 2015*; *Widloski and Foster, 2022*). The hippocampus and related structures, for example, entorhinal cortex (EC), are commonly known for containing cells that selectively fire for spatial features (*O'Keefe and Dostrovsky, 1971*; *Hafting et al., 2005*; *O'Keefe and Burgess, 1996*). We chose to represent the environment's structure with the default representation (DR) (*Piray and Daw, 2021*), which is a special case of the successor representation (*Dayan, 1993*). The latter has been proposed to explain firing properties of place and grid cells (*Stachenfeld et al., 2017*), and the DR gives an account for boundary cells in EC (*Piray and Daw, 2021*). Furthermore, we separated the representation of space into a general representation of 2D space and information about environmental barriers. It has been proposed that the hippocampal place cell code is formed by combining a learned representation of the task structure (e.g. space) with sensory information (*Whittington et al., 2020*). We propose that SFMA's similarity term is likely implemented through spatial representations found in the hippocampus and EC. Namely, place cell activity could provide the hippocampus access to the experience similarities or at least an estimate of them without the need for taxing computation.

The frequency of experience (*Kudrimoti et al., 1999*; *O'Neill et al., 2008*; *Gillespie et al., 2021*) as well as reward *Pfeiffer and Foster, 2013* have been shown to influence the content of replay. The experience strength term of SFMA is consistent with these experimental findings. Experience has also been incorporated in other models. For instance, the *Mattar and Daw, 2018* model uses the Successor Representation which depends on the agent's previous behavior. However, integrating the frequency of an experience $f_E(e)$ and the frequency/size of reward $f_R(e)$ into one variable, the experience strength, as SFMA does, renders replay less flexible and could negatively affect learning. For instance, extreme over-representation of the rewarded locations may slow down the propagation of reward information to locations far away from them. More flexibility could be introduced by separating the two parameters more clearly and computing experience strength as a weighted average $C(e) = w_E f_E(e) + w_R f_R(e)$. The weighting could then be flexibly adjusted depending on the situation. A potential way for a biological network to implement experience strength could be to increase the overall excitability of place cells after experience (*Pavlides and Winson, 1989*) and especially when reward was encountered (*Singer and Frank, 2009*).

Inhibition of return could easily be implemented via local inhibitory recurrent connections and experience strength through strengthening of connections between cell assemblies. While the main purpose of inhibition is that it enables the generation of sequences, it can be also viewed as a way of shifting priorities from experiences that were already used during learning to other experiences and, hence, ensuring the propagation of reward information. An interesting question pertains to whether a similar inhibition mechanism is utilized in the hippocampus. It was shown in modeling studies that the generation of sequences in a network does not strictly require inhibition (*Bayati et al., 2015*). However, forms of inhibition have been used to account for replay as well as preplay sequences in biologically plausible models (*Chenkov et al., 2017*; *Azizi et al., 2013*). We propose that the existence of inhibition implemented in our model could be tested by measuring the excitability of place cells following their reactivation during replay events.

In summary, the variables that SFMA uses for prioritization could be realized in an efficient manner by the hippocampus.

## The role of different replay modes in our model

While the default mode accounts for all varieties of replay statistics considered, it performs considerably worse during learning than the reverse mode since it does not produce reverse sequences as reliably. Reverse sequences are important for learning because they can propagate rewards information back to preceding actions that led to the reward (*Foster and Wilson, 2006*).

To benefit from the advantages of both replay modes, we propose that they could play different roles during learning. There is support in the literature for this view. *Foster and Wilson, 2006* found that reverse replay can occur after even a single experience, which is predominantly generated by the reverse mode. Furthermore, the frequency of reverse replay has been observed to increase in response to positive reward changes (*Ambrose et al., 2016*). Hence, the reverse mode might be used preferentially when the situation requires learning and the switch to the reverse mode may be triggered by neurons encoding reward prediction error (*Schultz et al., 1997*). By contrast, *Wikenheiser and Redish, 2013* reported both forward and reverse replay sequences during awake resting phases occurring in roughly equal proportions (for a whole session). In a familiar environment, *Diba and Buzsáki, 2007* found that replay sequences in forward order at the beginning of a trial prior to running on a linear track and in reverse order at the end of a trial after running. These results are more consistent with the default mode and more balanced replay direction in familiar environments.

What could the function of the default mode be, if it does not speed up learning much compared to random replay? A possibility could be a supportive function in the RL process. For instance, in the context of deep RL it is well known that multi-layer networks trained with sequences suffer from learning instabilities (*Mnih et al., 2015*). These instabilities are addressed by randomly replaying experiences to break the strong temporal correlations between sequential experiences. While not completely random, the default mode is more likely to break proper reverse and forward sequences. Another possible function that could be implemented by the default mode is the reactivation of experiences which otherwise would not (or only rarely) be reactivated in the reverse mode. Recent findings of pessimistic replay, i.e., the replay of sub-optimal experiences, in humans (*Eldar et al., 2020*) supports this view. Interestingly, follow-up modeling work (*Antonov et al., 2022*) suggests that optimal replay occurs predominantly during initial trials and then switches to pessimistic replay. However, this switch is dependent on the *utility* of experiences (*Mattar and Daw, 2018*). The default mode could be viewed as enabling these pessimistic replays, and the switch from optimal to pessimistic replay corresponds to the dynamic mode in our model.

Alternatively, the default mode might not be directly related to RL, but serve a different function altogether. For instance, the sequences generated in the default mode might be important for maintaining synaptic connections which otherwise would not be maintained due to the specific distribution of reverse replays (*Hebb, 1949*; *Caporale and Dan, 2008*). Another function often attributed to hippocampal replay is that of planning (*Diba and Buzsáki, 2007*; *Buckner, 2010*; *Buhry et al., 2011*) by simulating possible future trajectories to inform subsequent behavior. The default mode could, in principle, support this function. In contrast, since the reverse mode almost exclusively produces reverse sequences, it could not support the prospective sequences needed for planning.

Two further replay modes can be defined in our model, which we did not consider in our study. An attractor mode, which compares the next state of experience $e_t$ with the next state of all stored experiences e. Hence, this mode looks for experiences that end in the same state, so that neighboring experiences are activated in a ring-like fashion (*Figure 2—figure supplement 3*). Since we are not aware of any experimental studies reporting such ring-like replay, we did not consider this mode in our study. Another possible replay mode, is the forward mode, which compares the next state of experience $e_t$ with the current state of all stored experiences e. The forward mode exclusively produces forward replays, which are not efficient for learning, but could very well efficiently support planning. The latter is, however, outside the scope of the current study. Furthermore, the forward mode is not as flexible as the default mode in generating transitions between states that were never (or rarely) experienced, and replaying such transitions could be useful for discovering novel solutions.

How might default and reverse modes be implemented by a biological network? The default mode could be implemented as an attractor network where each experience's current state is stored as an attractor. The currently reactivated experience/state would be represented by a bump of activity in the network. Due to self-inhibition the bump would be driven to leave the current attractor state and move to a close-by, i.e., similar, one.

The reverse mode requires two attractor networks, net0 and net1, to store the current and subsequent states of each experience, respectively. They are assumed to use the identical representations. In addition, the state transitions $(s_t, s_{t+1})$ are encoded in the weight matrix $w_{10}$ between net1 and net0. An experienced transition is stored by learning a connection from $s_{t+1}$ in net1 to $s_t$ in net0. When an experience is replayed in the reverse mode, the pattern representing $s_{t+1}$ is activated in net1, which in turn excites the state $s_t$ in net0. The weights $w_{01}$ from net0 to net1 are assumed to be one-to-one connections, so that the activity pattern in net0 activates the most similar pattern in net1, equivalent to our algorithm finding, in all experiences, the next state that is closest to the current state of the reactivated experience. And the process of reactivating experiences iterates from here, thereby resulting in reverse replay.

Since the reverse replay has to perform one more computational step than the default mode, that is, the transition from $s_{t+1}$ to $s_t$, this would suggest that generating sequences in the reverse mode is slower, which is consistent with experimental observations (*Liu et al., 2021*).

## Current limitations of SFMA

SFMA accounts for the statistics of replay sequences and generates sequences that drive learning efficiently. However, in its current version, it addresses neither when replay events should occur nor the overall amount of replay. This limitation also extends to other similar models that implement a fixed number of update steps (*Mattar and Daw, 2018*) or initiate replay at predefined points in time (*Khamassi and Girard, 2020*). This limitation leaves certain aspects of a variety of experimental findings outside of the scope of our model. For instance, *Cheng and Frank, 2008* reported that replay decreases across sessions as the animals became more familiar with their environment, and *Ambrose et al., 2016* found that the number of SWRs and replay increases or decreases with changing reward magnitude. Nonetheless, *Mattar and Daw, 2018* argue that their model accounts for both these findings by focusing on the number of 'significant' replays produced by their model, which decreases with familiarity with the environment and when reward decreases, and increases when reward increases. For SFMA, a similar effect w.r.t to the familiarity occurs in the default mode: As familiarity with the environment increases and the agent's policy stabilizes, longer forward and reverse sequences are produced at the start and end of trials (*Figure 3—figure supplement 4*). Given the fixed number of replay steps, this results in a lower number sequences.

In PMA, the amount of reverse replay is driven by the gain term of the prioritization scheme. This is because gain is the main driver of reverse sequences and gain increases for positive reward changes. However, if gain is low, replay is dominated by the need term instead thereby resulting in the absence of reverse replays. While SFMA cannot account for the overall change in amount of replay due to changes in reward it can explain the shift in the statistics of occurring replay with the dynamic mode. The dynamic mode determines whether to produce replay using the reverse mode, which predominantly generates reverse sequences, given the recent history of prediction errors. Indeed, in open field and T-maze environments, we find that reward changes results in replays being generated in the reverse mode more often (*Figure 3—figure supplement 3*). However, this does not match the experimental findings which show a decrease of reverse replay for negative reward changes (*Ambrose et al., 2016*). Unlike PMA, SFMA therefore cannot account for the differential effects of reward change in its current form.

Recent findings show that replay can be influenced by motivational shifts (*Carey et al., 2019*). SFMA and other similar models may not be suitable to account for these findings, since they do not differentiate between different types of reward and do not consider different behavioral states like hunger or thirst. Changes in behavior due to motivational shifts could be explained by the learning of separate Q-functions for different sources of reward and then choosing the Q-function which corresponds to the current motivation. In *Carey et al., 2019* two reward sources, that is, food and water, were available to the animals. When one was restricted behavior shifted towards it, but replay shifted toward the unrestricted one. While SFMA may produce replays representing the unrestricted reward source, it would not be able to account for the observed shifts in replay content due to shifts in motivation. SFMA could potentially account for the shift in replay content by including the recency of experience during prioritization. If more remote experiences have their priority increased, the content of replay could shift in a similar way to the one reported. Another account for the content of replay shifting to represent the unrestricted reward source is given by the model by *Antonov et al., 2022*.

In their model, which is partly based on PMA, the agent's Q-function is subjected to forgetting, which eventually leads the utility of transitions not updated recently to increase. Since during behavior the restricted reward source is preferentially visited, the unrestricted reward source would be more affected by forgetting. This would lead to replay representing the unrestricted reward source more often. Both potential mechanisms would produce the shift of replay content as a result of the behavioral history rather than due to motivational shift directly.

*Gillespie et al., 2021* reported that awake replay preferentially represented previously rewarded goal locations rather than the current one. Since experience strength is modulated by both reward and previous behavior this finding could potentially be accounted for by our model. However, the experiment also found that locations that have not been visited recently are represented more often. This effect of behavioral recency can currently not be accounted for by SFMA. The two potential mechanisms discussed in the above paragraph could enable SFMA to also account for the effect of behavioral recency.

## Related work

The work closest to ours is the state-of-the-art PMA model by *Mattar and Daw, 2018*. It proposes that replay prioritizes experiences according to their utility, which in turn is defined as the product of gain and need. Gain represents the policy improvement that would result hypothetically if an experience were reactivated and need represents the expected future occupancy of the experience's current state. As mentioned in the introduction, it is not clear how a biological neural network could compute the utilities for stored experiences. Our model circumvents this problem by basing the probability of reactivation on behavioral and environmental variables and offers a potential account for how the utility of experiences could be approximated.

Furthermore, the implementation of PMA contains a few oddities. Because utility is defined as a product of gain and need, the gain must have a nonzero value to prevent all-zero utility values. This is achieved by applying a (small) minimum gain value. A problem arises for the special $n$-step update, which updates the Q-function for all n states along the replay trajectory, considered by their model for which gain is summed over $n$ steps and which is the main driver of forward sequences (especially, for offline replay). The minimum gain value is applied before summation, so that the gain artificially increases with sequence length. Furthermore, Mattar & Daw state that 'EVB ties are broken in favor of shorter sequences', where EVB stands for 'expected value of backup', that is, the utility. By contrast, due to their implementation, longer sequences are always favored in the case when gain is zero for all experiences, which can occur when the environment does not contain reward or the environment is highly familiar. This artifact could be avoided by applying the minimum value only once after the summation of gain values. In our hands, this reduces the model's ability to produce sequences during offline replay, because the utility is dominated by the gain term (*Figure 4—figure supplement 5*).

A different view of hippocampal replay was presented by *Khamassi and Girard, 2020*. They argue that replay can be viewed as a form of bidirectional search in which learning is facilitated by alternating between phases of prioritized sweeping (*Moore and Atkeson, 1993*), which prioritizes experiences according to how their reactivation affects behavior, and trajectory sampling (*Sutton and Barto, 2018*), which explores potential routes by simulating trajectories through the environment. In simulations, they show that bidirectional search performs better than prioritized sweeping and trajectory sampling alone. Importantly, they also were able to reproduce the occurrence of shortcut replays. Their model is, in principle, able to generate random walk replays due to the fact that, in the absence of reward, replay would be dominated by the trajectory sampling phase under a uniform action policy. However, their model does not give an account for how replay could adapt to environmental changes.

*McNamee et al., 2021* proposed a generative model of replay which adapts replay statistics depending on the cognitive function that it supports. Unlike the models discussed above, this model focuses on states, that is, locations, reactivated and ignores the combination of state and action. Their model produces random walks and they show that these replay statistics facilitate the learning of a successor representation of the environment. However, this model lacks a proper account for why replay should be able to quickly adapt to changes in the environment, and assumes a generator for (random) sequences which is handcrafted for each simulation. Furthermore, their model does not seem to allow for differentiating forward from reverse sequences.

## Conclusion

Our model gives a biologically plausible account of how the hippocampus could prioritize replay and produce a variety of different replay statistics, and fits well with contemporary views on replay. Furthermore and most importantly, the replay mechanism facilitates learning in goal-directed tasks, and performs close to the current state-of-the-art model without relying on computations of hypothetical network updates.

# Materials and methods
## Reinforcement learning

For our computational study we relied on reinforcement learning (RL) (*Sutton and Barto, 2018*) which makes it possible to learn optimized sequences of choices based on sparse rewards. In RL, an agent interacts with an environment and tries to maximize the expected cumulative reward (*Sutton and Barto, 2018*):

$$R = \mathbb{E}\left[\sum_t \gamma^t r_t\right], \tag{2}$$

where $\gamma$ is the so-called discount factor that weighs the importance of rewards that will be received in the future relative to immediate ones, and $r_t$ is a real-valued reward received at time $t$. The RL agent must learn to choose those actions that maximize the expected cumulative reward, the so-called optimal policy, through interactions with its environment. These interactions are generally formalized as experience tuples $e_t = (s_t, a_t, r_t, s_{t+1})$, where $s_t$ and $s_{t+1}$ denote the current state and next state, respectively. The action taken by the agent in state $s_t$ is represented by $a_t$ and the scalar reward received is represented by $r_t$. The critical problem in RL is crediting the right choices leading to a reward that was received only (much) later.

A popular algorithm that solves this learning problem is Q-learning (*Watkins, 1989*) which uses a state-action value function $Q(s, a)$, also called the Q-function, to represent the cumulative future reward expected for taking action $a$ in state $s$. The Q-function is updated from experiences in each time step $t$ by:

$$Q(s_t, a_t) \leftarrow Q(s_t, a_t) + \eta \left[r_t + \gamma \max_a Q(s_{t+1}, a) - Q(s_t, a_t)\right] \tag{3}$$

Here, $\eta$ denotes the learning rate. For all simulations the learning rate was set to $\eta = 0.9$. The term in brackets is known as the temporal difference error, and represents by how much the last action and potentially collected reward has changed the expected discounted future reward associated with the state. RL generally requires many interactions with the environment, which results in slow learning. To address this problem *Lin, 1992* introduced experience replay, where interactions with the environment are stored as experiences $e_t$ and reactivated for updating the Q-function later. In its simplest form, experience replay samples stored experiences uniformly at random. Learning from replay can be further improved by prioritizing experiences to be replayed (*Sutton and Barto, 2018*; *Schaul et al., 2016*).

## Spatial structure and frequency-weighted memory access (SFMA)

For our simulations, we employed the Dyna-Q architecture in combination with grid world environments in a similar fashion to *Mattar and Daw, 2018*. We view the reactivation of specific experiences as analogous to the reactivation of place cells in the hippocampus during awake and sleeping resting states. Like the original Dyna-Q architecture, our model stored experiences for all environmental transitions in a memory module. However, we stored additional variables for later prioritization of experiences during replay: The experience strength $C(e_i)$, the experience similarity $D(e_i|e_j)$ and inhibition of return $I(e_i)$. The experience strength is a count of how often a certain environmental transition was experienced. Following an experienced transition $e_i$ the experience strength was updated as follows:

$$C(e_i) \leftarrow C(e_i) + 1 \tag{4}$$

Additionally, experience strength was modulated by rewards experienced during behavior. If reward was encountered, the strength of experiences $e_i$ was increased weighted by their similarity to $e_r$ according to the following update rule:

$$C(e_i) \leftarrow C(e_i) + rD(e_i|e_r) \tag{5}$$

where $r$ is the scalar reward encountered and $e_r$ is the corresponding experience. The similarity between two experiences $e_i$ and $e_j$ was encoded by $D(e_i|e_j)$ and was based on the Default Representation (DR) (*Piray and Daw, 2021*). Specifically, we used the DR to represent open space and barrier information separately. The DR representing open space was computed directly from the state-state transition model $T$ which represents the default policy:

$$D = (X - \gamma_{DR}T)^{-1} \tag{6}$$

Here, $X$ is the identity matrix and $\gamma_{DR}$ is the DR's discount factor. The DR's discount factor was set to $\gamma_{DR} = 0.1$ unless otherwise stated. Barrier information was provided during initialization as a list of invalid transitions and could be updated in case of environmental changes. The low rank matrix update of the DR was computed for these invalid transitions. Inhibition $I(e_i)$ was used to prevent the repeated reactivation of the same experience and applied following reactivation of an experience to all experiences $e_i$ that share the same starting state $s_t$:

$$I(e_i) \leftarrow 1 \tag{7}$$

Inhibition terms decayed in each time step with a factor $\lambda$:

$$I(e_i) \leftarrow \lambda I(e_i) \tag{8}$$

Inhibition was reset between replay epochs and the decay factor was set to $\lambda = 0.9$ unless otherwise stated.

Replays were initiated in two different ways depending on whether they were simulated for awake resting states or sleep. Awake replay was initiated at the agent's current state, sleep replay was initiated in a randomly chosen state based on the relative experience strengths:

$$p(e) = \frac{C(e)}{\sum_i C(e_i)} \tag{9}$$

Given the most recently reactivated experience $e_t$, an experience $e$ was stochastically reactivated based on the priority ratings $R(e|e_t)$:

$$R(e|e_t) = C(e)D(e|e_t)[1 - I(e)] \tag{10}$$

The activation probabilities $P(e|e_t)$ were computed by applying a customized softmax function to the normalized priority ratings:

$$P(R_i) = \frac{e^{\beta_M R_i} - 1}{\sum_i \left(e^{\beta_M R_j} - 1\right)} \tag{11}$$

where $\beta_M$ is the inverse temperature with $\beta_M = 9$, unless otherwise stated. Since in the usual softmax function priority ratings of zero would yield non-zero reactivation probabilities, we subtracted one from each exponential in our customized softmax function. The replay epoch ended, that is, replay stopped, when either a maximum number $N$ of replayed experiences was reached or the priority ratings for all experiences were below a threshold of $\theta = 10^{-6}$.

Since each experience tuple contains two states, that is, the current and next state, that could be compared to one another to determine similarity of experiences $D(e|e_t)$, we explored two different replay modes. The default mode, which computes similarity based on the two current states, and the reverse mode, which compares the next state of experience $e$ to the current state of experience $e_t$ (*Figure 2B*). Additionally, we explored the possibility for the agent to dynamically decide which replay mode to use based on the cumulative temporal difference errors experienced since the last trial - we call this the dynamic mode. The dynamic mode defines the probability of generating replay in the reverse mode as:

$$P(reverse) = \frac{1}{1 + e^{-5\Delta - 2}} \tag{12}$$

where $\Delta$ is sum of absolute temporal difference errors that were encountered since the last trial.

**Table 1.** Training settings for each task.

| Environment | Trials | Steps per Trial | Replay Length |
| --- | --- | --- | --- |
| Linear Track | 20 | 100 | 10 |
| Open Field | 100 | 100 | 10 |
| Labyrinth | 100 | 300 | 50 |

## Model implementation

We implemented our model as well as all of our simulations and analysis code in Python 3 using the CoBeL-RL framework (**Walther et al., 2021**; **Diekmann et al., 2022**) (https://doi.org/10.5281/zenodo.5741291). The code for all simulations has been made available on Github: https://github.com/sencheng/-Mechanisms-and-Functions-of-Hippocampal-Replay (copy archived at **Diekmann, 2023**).

## Simulation: Navigation tasks

Spatial learning performance was tested in three grid world environments: a linear track of size 10 × 1 states, an open field of size 10 × 10 states and a labyrinth of size 10 × 10 states. Each environment had one specific goal, which yielded a reward of 1, and one starting state. We trained agents which use SFMA in default, reverse, and dynamic modes. The discount factor used for learning was

$\gamma = 0.99$. Simulations were repeated with different discount factors $\gamma_{DR} = \{0.1, 0.2, ..., 0.9\}$ for the DR

and their performance was averaged. In the last mode, the agent dynamically chose either the default or reverse mode based on the cumulative temporal-difference error experienced in the current trial. The higher the cumulative temporal-difference error was, the higher was the probability of choosing the reverse replay mode. For comparison, we also trained agents that learned from online experience only, with random replay, and Prioritized Memory Access (**Mattar and Daw, 2018**) (PMA). To account for the different sizes and structure of the environments, the number of steps per trial as well as the length of replays were different for each environment (**Table 1**). Since the model by **Mattar and Daw, 2018** uses additional replay at the start of each trial, we split the replay length in half. Furthermore, during action selection we masked actions which would lead the agent to same state. Reasons for this were twofold: First, by masking action we sped up learning for all agents by removing pointless actions. Second, it mitigated the unfair advantage of PMA which in its prioritization ignores experiences that lead into the same state. To test the optimality of SFMA's different replay modes we computed for each step in each replay whether the reactivated experience was optimal, that is, it had the highest utility as defined by **Mattar and Daw, 2018**. For simplicity we assumed uniform need as in **Antonov et al., 2022**.

We further analyzed the directionality of replay in the default and reverse modes depending on when, in a trial, it was initiated, that is, at the start and end of a trial. Simulations were repeated with the same training settings listed above except that the number of trials for the linear track was extended to 100. Furthermore, additional replays were generated at the start and end of each trial but they did not contribute to learning. The directionality of these additional replays was measured by the difference between the number of forward and reverse pairs of consecutively reactivated experiences $e_{t-1}$ and $e_t$. A pair $e_{t-1}$ and $e_t$ was considered to be forward when the next state of $e_{t-1}$ was the current state of $e_t$, reverse when the current state of $e_{t-1}$ was the next state of $e_t$ and unordered otherwise.

## Simulation: Random walk replay

To replicate the experiment by **Stella et al., 2019** we chose an open-field grid world with 100 × 100 states. The large size of this environment allowed us to simulate replays of different lengths. We considered two cases: Homogeneous experience strengths and heterogeneous experience strengths, which were highest at the center of the environment and decreased with distance to half the maximum value at the borders. In each simulation a total of 50 replays of length 500 were generated. At such a large length, replays can easily reach the environment's borders if initialized near the borders. We

therefore initialized replays exclusively at the environment's center. Simulations were repeated for different values of DR discount factor $\gamma_{DR}$ and inhibition decay $\lambda$:

$$\gamma_{DR} \in \{0.01, 0.1, 0.2, 0.3, 0.4, 0.5, 0.6, 0.7, 0.8, 0.9\} \tag{13}$$

$$\lambda \in \{0.00, 0.1, 0.2, 0.3, 0.4, 0.5, 0.6, 0.7, 0.8, 0.9\} \tag{14}$$

We applied the Brownian Diffusion Analysis used by *Stella et al., 2019* to the output of our simulations. Furthermore, we tested the randomness of replay starting positions as well as their directions. We did so by initializing 1000 replays and restricted replay length to five. Values for the DR discount factor and inhibition decay were fixed at $\gamma_{DR} = 0.9$ and $\lambda = 0.9$, respectively. The distribution of starting positions was computed by counting how often a given state occurred as the current state of initially replayed experiences. For the distribution of initial replay directions we first divided the environment into spatial bins of size 20 by 20 states. Direction was divided into 90° bins, i.e., bins centered at 0°, 90°, 180° and 270°. For each spatial bin we then counted how often replay direction fell into respective direction bins. Lastly, we computed displacement distributions at different time steps, $\Delta t = 1, 2, 3, 4$, relative to initially replayed positions in a $25 \times 25$ states window. This was done by averaging the replayed position relative to the start of replay.

## Simulation: Shortcut replay

For this simulation, we simplified the maze used by *Gupta et al., 2010* as an $11 \times 7$ states figure-eight-maze-like grid world. Reward locations on either side were reduced from two to one location. Since our main interest lay in the effect of specific stereotypical running behavior expressed by the animals on replay, basic running patterns were predefined:

1. From starting position to left side's reward location.
2. From starting position to right side's reward location.
3. From left side's reward location to left side's reward location.
4. From right side's reward location to right side's reward location.
5. From left side's reward location to right side's reward location.
6. From right side's reward location to left side's reward location.

From these running patterns, we created sessions consisting of 20 trials. Each session was characterized by one pattern during the first 10 trials, which could change to a different pattern for the remaining 10 trials, for example, 10 right laps followed by 10 left laps. Running patterns 1 and 2 were only used in the initial trial of a session. Patterns 3 and 4 were used when the agent was supposed to run laps on the same side, and patterns 5 and 6 were used when the agent was supposed to alternate between the two reward locations. We considered the following pattern combinations (used patterns are listed in parenthesis ordered numerically):

- Right-Left (from patterns 2, 3, 4, and 6).
- Right-Alternating (from patterns 2, 4, 5, and 6).
- Alternating-Left (from patterns 2, 3, 5, and 6).
- Alternating-Alternating (Control Condition; from patterns 2, 4, 5, and 6).

After each trial, 200 replays were generated and stored for later analysis. Since reactivation probabilities are controlled by the softmax function's inverse temperature parameter $\beta_M$, we also repeated each simulation for a range of values $\beta_M \in \{5, 6, ..., 15\}$. To detect shortcut replays, we first generated state sequence templates for each corner of the maze (i.e. TL, TR, BL and BR) in forward and backward direction. A template was considered a match if the number of mismatches was at most two. If two corners that together would form a shortcut (e.g. TL in backward direction and TR in forward direction) were detected in a replay and were separated by $8 \pm 2$ states they were considered to form a shortcut. The different possible shortcuts were:

- From TL (backward) to TR (forward).
- From TR (backward) to TL (forward).
- From BL (forward) to BR (backward).
- From BR (forward) to BL (backward).

For our additional learning simulations, an RL agent was trained in the environment. Since a simple RL agent, like the one employed, is not capable of expressing the same stereotypical running behavior,

we restricted selectable actions such that the agent can only run forward. The only relevant choice was therefore limited to the maze's decision point. If the agent picked the correct choice it received a reward of 1 for that trial. Furthermore, since the agent is not capable of learning in conditions with alternating reward arms for lack of working memory, we restricted simulations to the Right-Left condition. Simulations were repeated for agents trained with SFMA in default and reverse mode as well as random replay. To better differentiate the effect of replay type, we allowed but one replay after each trial and increased the number of trials per session to 200. The discount factor used for learning was $\gamma = 0.99$.

## Simulation: Adaptive replay

To evaluate our model's ability to adapt to structural changes in an environment we set up three structurally different environments and presented them successively to our model. Replays were simulated in each environment and analyzed by computing the fraction of invalid transitions. Invalid transitions were defined as two consecutively replayed experiences whose current states were separated by a barrier. To evaluate the importance of the representation chosen for experience similarity, we performed all simulations using the Euclidean distance for the experience similarity, that is,

$$D(e_i|e_j) = \exp\left(-\|s_i - s_j\|\right) \tag{15}$$

## Simulation: Preplay

We simplified the experimental setup from *Ólafsdóttir et al., 2015* to a 5 × 7 T-maze-like grid world. A barrier was inserted to separate stem from arms. The simulation was divided into three phases. First, in the run phase the agent was made to run up and down the stem, and rewards were provided at the two ends of the stem. Second, in the cue phase we replicated the attention paid to the arms by the animals as an increase of experience strengths at the arms. Visual exploration was associated with a lower increase in experience strengths than physical exploration by a factor we call attention strength. The values focused on for the strength of attention was:

$$\{0.0, 0.02, 0.04, 0.06, 0.08, 0.1, 0.12, 0.14, 0.16, 0.18, 0.2\} \tag{16}$$

The right arm was chosen arbitrarily as the cued arm that contained a reward and therefore was allocated more attention than the other, uncued arm (asymmetric allocation of attention). For the asymmetric allocation of attention, we focused on values greater than 0.5 as they best reflect the attention reported:

$$\{0.5, 0.55, 0.6, 0.65, 0.7, 0.75, 0.8, 0.85, 0.9, 1.0\} \tag{17}$$

The increase of experience strengths for cued was computed by multiplying attention strength and asymmetry with the increase that would be expected if the agent actually traversed the cued arm. The increase of experience strengths for the uncued arm was computed similarly by using inverse symmetry, i.e., $1 - symmetry$. For each combination of values for attention strength and asymmetric allocation of attention we simulated 5000 preplays of length six. Preplay sequences were identified with template matching. Templates were prepared for stem, cued, and uncued arms for forward and reverse sequences. In a final choice phase, the value combination (0.75 and 0.14 for asymmetric allocation of attention and attention strength, respectively) which yielded a preplay fraction closest to the one reported by *Ólafsdóttir et al., 2015*, i.e., 7.37, was used to train the agent's Q-function. The discount factor used for learning was $\gamma = 0.99$. The choice at the decision point was compared between before and after training assuming a softmax action selection policy with an inverse temperature parameter of $\beta = 20$. The action that leads into a wall and the action that leads back into the stem were ignored since we wanted to focus on the agent's preference for the cued vs the uncued arm.

## Simulation: Reward modulation

Simulations were run in an open field environment of size 10 × 10 states, and a T-maze environment with a stem length of five states and an arm length three states. Reward of magnitude one was located in the left upper corner in the open field environment and the right arm in the T-maze environment. Agents were trained for 100 trials each of which lasted at most 100 steps. Online replays of length 10 were generated at the begin and end of trials. Additionally, offline replays with same length were

also generated. However, only replays generated at the end of trials were used to train the agent. The reward modulation of experience strength, that is, the increase in experience strength due to reward, had a value of either 1 or 10. Simulations were repeated 100 times for each combination of replay mode (i.e. default, reverse, and dynamic) and value for reward modulation. The representation of replay locations was analyzed by computing their reactivation probability across all replays separately for trial begin, trial end, and offline replays.

## Simulation: Aversive shock zone

To replicate the findings from *Wu et al., 2017* we used a linear track gridworld environment with a length of 20 states. The first two states from the left represented the shock zone. Dark and light zones of the environment were represented by the left and right halves, respectively. Experience with the environment preceding shock (i.e. during Run 1, Run 2 and Pre) was simulated by making the agent run 30 laps on the linear track and then multiplying the experience strengths of dark and light zones with their reported occupancy (i.e. 74% and 26%). Finally, ten trials were dedicated to Shock and Post phases. For the initial shock trial (Shock), the agent was made to run until the shock zone where it received a shock of magnitude 1. For the remaining nine trials (Post) the agent was running laps in the light zone of the environment. Twenty replays of length ten were generated for each trial in Shock and Post phases. For the shock phase, replays were generated following shock while for the Post phase replays were generated when the agent stopped in front of the dark zone. Simulations were repeated 100 times and the reactivation probability for each state was computed. Additionally, we analyzed how the generated replays could result in shock zone avoiding behavior by inspecting the agent's Q-function. More precisely, for each state we computed the probability of the action pointing away from the shock zone being the better option (i.e. the action is associated with less shock). The learned Q-functions were analyzed separately for the first and second half of the Post phase. To account for the fact that the Q-learning update cannot propagate information about worse rewards due to the max operator we repurposed the Q-function to represent the (positive-valued) discounted cumulative punishment instead of reward. The Q-function was not updated during behavior to focus on the replays' effect on behavior.

## Additional information

### Funding

| Funder | Grant reference number | Author |
|---|---|---|
| Deutsche Forschungsgemeinschaft | 419037518 - FOR 2812 P2 | Sen Cheng |

The funders had no role in study design, data collection and interpretation, or the decision to submit the work for publication.

### Author contributions

Nicolas Diekmann, Software, Formal analysis, Investigation, Visualization, Methodology, Writing – original draft, Writing – review and editing; Sen Cheng, Conceptualization, Supervision, Funding acquisition, Writing – original draft, Project administration, Writing – review and editing

### Author ORCIDs

Nicolas Diekmann (iD) http://orcid.org/0000-0003-3638-7617
Sen Cheng (iD) http://orcid.org/0000-0002-6719-8029

### Decision letter and Author response

Decision letter https://doi.org/10.7554/eLife.82301.sa1
Author response https://doi.org/10.7554/eLife.82301.sa2

## Additional files

### Supplementary files
• MDAR checklist

### Data availability
The current manuscript is a computational study, so no data have been generated for this manuscript. Modelling code has been made publicly available at https://github.com/sencheng/-Mechanisms-and-Functions-of-Hippocampal-Replay (copy archived at *Diekmann, 2023*).

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
