## [Editor Report]

This paper proposes a new, biologically realistic, computational model for the phenomenon of hippocampal replay. This is an important study with relevance for a broad audience in neuroscience. The proposed model convincingly simulates various aspects of experimental data discovered in the past.

---

## [Decision Letter]

**Decision letter after peer review:**

Thank you for submitting your article "A Model of Hippocampal Replay Driven by Experience and Environmental Structure Facilitates Spatial Learning" for consideration by *eLife*. Your article has been reviewed by 3 peer reviewers, and the evaluation has been overseen by a Guest Reviewing Editor and Laura Colgin as the Senior Editor. The following individual involved in review of your submission has agreed to reveal their identity: H Freyja Ólafsdóttir (Reviewer #2).

Essential revisions:

Reviewers agreed that this is an important study with relevance for a broad audience in neuroscience. Reviewers specifically commended the simplicity and elegance of the proposed computational model and the clarity of the manuscript. There were, however, also several concerns that we would ask you to address in a revision.

1) All reviewers agreed that there should be further simulation analyses to address several seminal empirical data in the field of replay. Of particular importance are those studies that are simulated in Mattar and Daw 2018 paper. Is the model able to recapitulate reward over-representation in the open field and T-maze (Mattar and Daw's Figure 4). Another crucial experiment (also simulated in the Mattar and Daw 2018) is the phenomenon of non-local replay. Reviewers agree that it's important that computational models of replay capture these characteristic aspects of replay.

2) Reviewers also agreed that it is important that authors perform further simulation analyses to address other experiments simulated in Mattar and Daw 2018. These include responsiveness of reverse replay to reward change, replay counterintuitively entering an aversive zone, and the effect of experience on replay frequency. Even if the proposed model cannot account for all of these findings, it is important that authors transparently report them (maybe with supplementary figures) and discuss them extensively.

3) The fact that the model has two modes (default and reverse) to account for empirical findings is also a major issue. Reviewers agree that the underlying mechanism regarding these two modes is not clear. Do they work in parallel? or is there a dynamic switch between the two? How does the dynamic switch work computationally? And how is this aspect of the model related to experimental data? Please see specific comments by Reviewers 2 and 3 about this issue.

4) In addition, individual reviewers made a number of suggestions to improve the manuscript. Please carefully consider these additional points when preparing the revised manuscript.

*Reviewer #1 (Recommendations for the authors):*

My point is especially important in relation to the model by MandD. For example, does the current model explain evidence that replay over-represents reward locations (e.g. Figure 4 in MandD)? I suggest conducting additional simulation analysis, particularly for the experimental data explicitly presented in MandD and discussing any of those experiments that cannot be explained by the current model (if any).

*Reviewer #2 (Recommendations for the authors):*

In their paper, Diekmann and Cheng describe a model for replay generation that is able to replicate multiple seminal findings in the field. Specifically, in their model experiences get selected for replay depending on their strength (which is in turn determined by their frequency of occurrence and reward), their similarity to other experiences and inhibition that prohibits experiences that match the current one too much from being reactivated (which leads to the generation of replay sequences). Further, they adopt two different methods for computing similarity which they show produces complementary results. With this rather simple and intuitive model Diekmann and Cheng are able to re-produce important, but divergent, findings in the field. For example, why replay switches directionality at different time points on a maze, how it can generate synthetic experiences, respects boundaries, etc. Being able to account for these different findings with a single mechanism is an important goal and no small feat.

However, I still have some reservations about how much the current model differentiates itself – and represents an improvement – of a recently published model (the so-called prioritized memory access (PMA) model for replay [33]). Further, although their model provides a relatively comprehensive account of multiple replay findings, it doesn't address many studies that show awake replay is non-local (even depicting experiences in remote environments). Finally, I would like the authors to elaborate on how their 'default' and 'reverse' policy could be implemented biologically and how these two modes relates to current thinking in the replay field. I elaborate on my points below:

– Benefit over PMA model: The authors maintain that their model represents a significant improvement from the PMA model as the latter is not biologically feasible. In the PMA model, replay gets selected based on the utility of a possible experience reactivation which is partly derived from its "gain" term. Here, the authors' critique is that if the calculation of gain requires access to all stored experiences, this becomes very computationally laborious and biologically unfeasible. Yet, it's not clear to me in the current model's implementation that it represents a clear advantage in this way. Namely, to compute similarity – which influences replay prioritization – the current experience state needs to be compared with all other experience states (or tuples). Thus, does their model not also require access to all stored experiences? If not, I think the authors need to spell out a bit more about how this still makes their model more efficient and biologically implementable than the PMA model. This is a particularly important point as one of the main 'selling points' of the paper is that it achieves what the PMA model achieves but in a physiologically realistic way.

– The authors show the model does well in replicating multiple replay findings in the field. However, they focus primarily on awake replay findings that show some relationship to current behaviour- e.g. depicting short cuts to goal locations, activating paths just taken, etc. However, many replay studies show that replay is very often non-local (e.g. Karlsson and Frank (2009); Olafsdottir et al. (2017)). How can their model accommodate these findings?

– In the current model there are two modes (policies) for computing experience similarity described; the default and reverse mode. The authors show you need both to effectively account for different replay phenomena (e.g. you need reverse for reward learning, particularly in environments where an animal's path is not prescribed). But what determines which policy is used? How would switching between the two be implemented in neural networks? Or do they propose they work in parallel?

– The default and reverse mode resonate with some contemporary theories of replay. Namely, a commonly held view is that awake replay may support planning/memory retrieval etc. whereas replay during rest periods supports memory consolidation. Perhaps the authors can elaborate on how their different modes, which resemble somewhat the planning vs consolidation mode, relate to these theories. Could it be that the default mode may be a sort of a planning mode – supporting on-going behaviour – whereas the reverse mode is more about learning?

*Reviewer #3 (Recommendations for the authors):*

In multiple instances the authors refer to the biological plausibility of their model, and the lack thereof in the model by Mattar and Daw. Intuitively, I agree with the authors. However, to my knowledge we don't know what is biologically plausible and what is not. I suggest some rewording to emphasize that their model is simpler and less computationally taxing compared to Mattar and Daw. Further, I would appreciate a more explicit discussion of the biological plausibility of their model. They get into this to some degree in the discussion, particularly with regards to the return of inhibition term (i.e., line 401). However, the description of how experience strength or distance could be computed could certainly be expanded beyond just referencing other studies. Importantly, the model occurs in two modes. How might the hippocampus gate these two modes?

In general I found the methods to be exquisitely articulated and comprehensive. However, the description of how replay is generated in their model could be expanded/clarified in the main text:

– I understood that an experience consists of a transition between two states (in addition to reward/action)- and that the entire experience (including both states) is reactivated during replay. However I think a supplementary figure showing how exactly one gets from the last panel of figure 2A to an ordered set of reactivated states as in Figure 5c could be helpful.

– Most importantly- I think it would be very useful for the authors to show more examples of replay events generated by their model, particularly with respect to Figure 3. For example, they report the directionality of replays in the open field/labyrinth, but I would like to know where the replays actually started/ended, and how this changed over trials. This would help clarify how the replays are behaviorally useful to the agent.

– The authors could better describe the logic/motivation of the default and reverse states. Initially I assumed that the default mode simply drives forward replay, and the reverse mode reverse replay- but this is not entirely the case. Why did the authors use just 2 modes (the first comparing the similarity between experience 1-state 1 and experience 2 state-1, and the second comparing the similarity between experience 1-state 1 and experience 2 state-2)- why not, for example, have other modes comparing experience 1-state 2 and experience 2 state-1, or experience 1-state 2 and experience 2 state-2? Many readers would benefit from a bit more intuition in terms of linking these modes to biological phenomenon.

– Can a single replay consist of both forward and reverse transitions? As mentioned above, a panel of example replays upfront could be very useful.

– Please confirm that only one map was constructed for linear environments. What was the reason for this choice, and would it impact the results to use two maps (one for each running direction, as in Mattar and Daw, and consistent with hippocampal fields on a linear track)?

– A description of how the default versus reverse modes are selected in the dynamic mode is provided in the methods, but should also be described in the main text.

– The definition of a replay epoch is in the methods but might be moved to the main text.

Can this model account for a broader scope of replay phenomena including: the sensitivity of reverse replay to reward magnitude or reward prediction error, over-representation of reward locations, or the tendency of forward replays to counter-intuitively lead towards an aversive (negatively rewarding) locations? As Mattar and Daw demonstrated that their model produces patterns of replay that match these arguably fundamental empirical findings, examining the fully overlapping set of scenarios would allow the models to be compared head-to-head. It is unclear whether either model can account for the recent observations that replay is enriched for locations not visited recently (Gillespie et al., 2021) or trajectories directed away from the animal's preferred goal location (Carey et al., 2019). I do not mean to suggest that the authors needs to perform all of the above tests. However, it would be of interest to at least include discussion about which empirical findings the current model could likely not account for.

Line 160: 'in our simulations the reverse mode produces more updates that were optimal than did the default mode'. Please define an optimal (policy?) update.

Line 165: I'm not sure what the authors mean by 'mixed' replays. Are they describing replays that traverse in one direction and then switch to the other (i.e., reverse to forward), as in Wu and Foster 2014?. If not, a better word choice might be 'interspersed'.

Line 188: Are there any statistical tests to accompany the statement that these log-log plots are linear?

Line 223: It is not clear to me what the alternating-alternating condition controls for.

Line 234: The explanation about the observed differences in short-cut replays under the different behavioral paradigms was somewhat confusing. A figure showing the relative amount of experience in each segment of the maze in each paradigm could help clarify.

Line 248: While it is interesting to ask whether the replay generated by your model facilitates learning in the task, I think it would be of broader interest and more informative to show specifically which replays are useful. For example, what was the distribution of replays that the default and reverse mode yielded on that task? How does task performance change if you limit replay to a subset, such as only reverse replays starting at the goal, or, most relevant here- only-short cut replays?

Figure 6A: What are reactivation maps? I.e., is this a summary or average of all reactivations, or an example of an individual replay epoch (I assume it's the former, but that should be specified). Individual replay examples would also useful.

Line 329: 'General representations of task structure in EC have been proposed to be forming hippocampal place cell code together with sensory information'- I suspect there was a typo in this sentence; I was unable to decipher its meaning.

Line 409: 'We argue that the minimum value should be applied after the summation of gain values" – what is the argument for applying the minimum value after summation? In general I wonder if the description of PMA's flaws (starting at line 404) should be moved to the related Supplemental Figure legend. I think more detail would be needed for most readers to understand the apparent error (for example, what is the 'special n-step update', what does it mean to apply before summation or after summation, and what makes one version better than the other)?

Figure 5e: Please state in the figure legend what happened at trial 100. Also, as noted elsewhere, I think it would be useful to compare performance with the dynamic mode (and possibly the PMA model).

Figure 5/Supplementary Figure 7: I'm confused as to why in Figure 5 there is little difference between the forward and reverse modes in the initial learning of the task, but a difference after the reward location switch, whereas in Supplementary Figure 7 the exact opposite pattern is shown on the T-Maze. In general, what explains the differences in the initial learning rate and the learning rate after the reward switch, especially on the T-maze? I don't see what information the agent learned in the initial phase that remained useful after the switch. Perhaps showing examples of the agents behavioral trajectory/decisions at different stages of the task would help.

Supplementary Figure 3: what is the take-home message of showing the displacement distributions with different inverse temperature parameters? Is the point that the results hold when using different temperatures? If so, why not provide a more quantitative metric, like the log-log plots using different inverse temperature parameters?

Supplementary Figure 8: The main text indicates that this figure was generated using the reverse mode, but the legend doesn't say so.

Methods line 463: 'The term in brackets is known as the temporal difference error'. This could perhaps be explained a bit more.

---

## [Author Response]

Essential revisions:Reviewers agreed that this is an important study with relevance for a broad audience in neuroscience. Reviewers specifically commended the simplicity and elegance of the proposed computational model and the clarity of the manuscript. There were, however, also several concerns that we would ask you to address in a revision.1) All reviewers agreed that there should be further simulation analyses to address several seminal empirical data in the field of replay. Of particular importance are those studies that are simulated in Mattar and Daw 2018 paper. Is the model able to recapitulate reward over-representation in the open field and T-maze (Mattar and Daw's Figure 4).

We agree that the over-representation of rewarded locations is an important feature of replay that we did not sufficiently address in our previous simulations and manuscript. In fact, reward always was an important factor in our model in that it modulates the experience strength variable. To demonstrate the effect of reward explicitly, we performed simulations in open field and T-maze environments and analyzed the representation of environmental states during replay. A corresponding section detailing the simulation and discussing the results was included along with a new figure (Figure 8, “Reward Modulation of Experience Strength Leads to Over-Representation of Rewarded Locations”).

To summarize our results, replay produced by our model can account for the over-representation of rewarded locations in both environments. Over-representation depends on the relative influence of reward and experience as well as where and how replay is initiated. If experience strengths are more strongly modulated by reward and replay occurs at the end of a trial or in an offline manner, rewarded locations are strongly over-represented (Figure 8C, D).

Another crucial experiment (also simulated in the Mattar and Daw 2018) is the phenomenon of non-local replay. Reviewers agree that it's important that computational models of replay capture these characteristic aspects of replay.

In our previous simulations of awake replay we initialized replay at the agent’s current location as a very simple way of modeling the bias of replay to start at an animal’s current location (Davidson et al., 2009). This is, of course, rather artificial and also does not allow for non-local awake replay, as the reviewers rightly pointed out. By contrast, since offline replay during sleep appears to be less biased by the animal’s current location, we stochastically initialized sleep replay in our model. By applying the same stochastic initialization to online replay, our model also produces non-local replays. We tested the occurrence of non-local awake replays in our simulation of Gupta et al. (2010) in the figure-8 maze as well as in an open field environment with uniform experience strengths (Figure 2—figure supplement 2A, B). Replay exhibited non-local replays with a preference for rewarded and often visited locations in both environments, but a bias for the current location only emerged in the figure-8 maze (Figure 2—figure supplement 2A, B). That bias can be reintroduced by increasing priority ratings at the current location (Figure 2—figure supplement 2CF). We expanded the section of the main text which introduces the initialization of replay and added a supplementary figure (Figure 2—figure supplement 2) demonstrating how our model can also account for non-local awake replays.

2) Reviewers also agreed that it is important that authors perform further simulation analyses to address other experiments simulated in Mattar and Daw 2018. These include responsiveness of reverse replay to reward change, replay counterintuitively entering an aversive zone, and the effect of experience on replay frequency. Even if the proposed model cannot account for all of these findings, it is important that authors transparently report them (maybe with supplementary figures) and discuss them extensively.

Our model focuses on the statistics of replayed locations when replay occurs and does not have a mechanism for determining the frequency and duration of replay events. Hence, our model produces replay at fixed times and with a fixed number of replay steps. This places some aspects of the studies mentioned by the reviewers out of the scope of our model. In principle, this also holds for other similar models, including the one by Mattar and Daw (2018). Nonetheless, our model may still account for some aspects of the findings. We added a new section in the discussion which addresses the current limitations of our model in more detail (“Current Limitations of SFMA”).

The responsiveness of reverse replay to reward change (Ambrose et al., 2016) can in part be accounted for by SFMA’s dynamic mode. We ran simulations in open field and T-maze environments and induced positive and negative reward changes during learning (Figure 3—figure supplement 3). The dynamic mode determines the probability of activating the reverse mode based on the recent history of temporal difference errors. Therefore, our model produces more reverse replays whenever there are prediction errors regardless whether the reward changes are positive and negative. Ambrose et al. indeed find the increase in reverse replay when rewards are increased, but, in contrast to our modeling results, reverse replay decreases when rewards are decreased. Ambrose et al. find very few SWRs. Neither our model nor the one by Mattar and Daw can explain this result, since both use a fixed number of replay steps in their model. However, they analyze the number of “significant” replays and that number decreases with reduced reward due to the resulting absence of gain, and gain is the sole driver of reverse sequences in their model.

Our model can account for replays representing paths into an aversive zone. The only requirement is some previous exposure to the environment. We included an additional simulation of the experiments by Wu et al. (2017) in the main text demonstrating that the (biased) experience is sufficient to explain replay entering the aversive zone (Figure 9A). We further show that the generated replays can support the learning of shock-avoiding behavior (Figure 9B).

We consider the decrease in the amount of replay across sessions (Cheng and Frank, 2008) to be outside the scope of our model (see above). Mattar and Daw also use a fixed number of replay steps in their model. However, they analyze the number of “significant” replays and that number decreases with familiarity. We actually find a similar effect for replay generated with the default mode. During early learning stages our model produces multiple short sequences. With growing experience this number decreases but the sequence length increases (Figure 3—figure supplement 1, Figure 3—figure supplement 2, Figure 3—figure supplement 4).

3) The fact that the model has two modes (default and reverse) to account for empirical findings is also a major issue. Reviewers agree that the underlying mechanism regarding these two modes is not clear. Do they work in parallel? or is there a dynamic switch between the two? How does the dynamic switch work computationally? And how is this aspect of the model related to experimental data? Please see specific comments by Reviewers 2 and 3 about this issue.

We acknowledge that the need for and the implementation of the two modes of our model were not made clear enough in the manuscript. In SFMA’s dynamic mode, replay is generated in either default or reverse mode depending on the recent history of temporal difference errors. To be more precise, the more temporal difference errors were encountered, the higher the probability is for activating the reverse mode. This will be the case during learning, when the an environment is novel (Figure 3) or when the reward changes (Figure 3—figure supplement 3). This error-signal-induced switch is consistent with the increase in reverse replay due to positive reward changes (Ambrose et al., 2016). We clarified the description of the dynamic mode and its connection to experimental findings in the manuscript.

How might default and reverse modes be implemented by a biological network? The default mode could be implemented as an attractor network where each experience’s current state is stored as an attractor. The currently reactivated experience/state would be represented by a bump of activity in the network. Due to self-inhibition the bump would be driven to leave the current attractor state and move to a close-by (i.e., similar) one.

The reverse mode requires two attractor networks, net0 and net1, to store the current and subsequent states of each experience, respectively. They are assumed to use the identical representations. In addition, the state transitions (s_t_, s_t+1_) are encoded in the weight matrix w_10_ between net1 and net0. An experienced transition is stored by learning a connection from s_t+1_ in net1 to s_t_ in net0. When an experience is replayed in the reverse mode, the pattern representing s_t+1_ is activated in net1, which in turn excites the state s_t_ in net0. The weights w_01_ from net0 to net1 are assumed to be one-to-one connections, so that the activity pattern in net0 activates the most similar pattern in net1, equivalent to our algorithm finding, in all experiences, the next state that is closest to the current state of the reactivated experience. And the process of reactivating experiences iterates from here, thereby resulting in reverse replay.

Since the reverse replay has to perform one more computational step than the default mode, i.e., the transition from s_t+1_ to s_t_, this would suggest that generating sequence in the reverse mode is slower, which is consistent with experimental observations (Liu et al., 2021).

We have added these clarifications to the manuscript.

4) In addition, individual reviewers made a number of suggestions to improve the manuscript. Please carefully consider these additional points when preparing the revised manuscript.

Further Manuscript Changes

We added a link to the Github repository with the simulation code in the Methods section:

“The code for all simulations has been made available on Github:

https://github.com/sencheng/-Mechanisms-and-Functions-of-Hippocampal-Replay."

We added details missing in the subsection “Simulation: Navigation Task”:

“Simulations were repeated with different discount factors = {0.1, 0.2, …, 0.9} for the DR and their performance was averaged.”

It was pointed out that our critique of Mattar and Daw (2018) could be interpreted as overtly aggressive/negative. This was not our intention. We therefore rephrased our points of critique in a more neutral manner. Furthermore, rather than opposing the Mattar and Daw model we position SFMA as a complementary model, which gives an account for how the utility-based prioritization could be approximated by hippocampus.

We shortened the legend text of Figure 3 since it was rather lengthy and the same information is conveyed in both the main text as well as legend text of Figure 3—figure supplement 1.

We added the author contributions.

The enumeration of supplementary figures changed because new ones were added:

Supplementary Figure 1 → Figure 2—figure supplement 1Supplementary Figure 2 → Figure 2—figure supplement 2Supplementary Figure 3 → Figure 3—figure supplement 1Supplementary Figure 4 → Figure 3—figure supplement 2Supplementary Figure 5 → Figure 3—figure supplement 3Supplementary Figure 6 → Figure 4—figure supplement 1Supplementary Figure 7 → Figure 4—figure supplement 2Supplementary Figure 8 → Figure 4—figure supplement 3Supplementary Figure 9 → Figure 4—figure supplement 4Supplementary Figure 10 → Figure 4—figure supplement 5

Reviewer #1 (Recommendations for the authors):My point is especially important in relation to the model by MandD. For example, does the current model explain evidence that replay over-represents reward locations (e.g. Figure 4 in MandD)? I suggest conducting additional simulation analysis, particularly for the experimental data explicitly presented in MandD and discussing any of those experiments that cannot be explained by the current model (if any).

The suggested simulations and analyses were added along with an additional section addressing our model’s current limitations in the discussion. We address the reviewer’s recommendation in more detail in our reply to the essential revisions 1 and 2.

Reviewer #2 (Recommendations for the authors):In their paper, Diekmann and Cheng describe a model for replay generation that is able to replicate multiple seminal findings in the field. Specifically, in their model experiences get selected for replay depending on their strength (which is in turn determined by their frequency of occurrence and reward), their similarity to other experiences and inhibition that prohibits experiences that match the current one too much from being reactivated (which leads to the generation of replay sequences). Further, they adopt two different methods for computing similarity which they show produces complementary results. With this rather simple and intuitive model Diekmann and Cheng are able to re-produce important, but divergent, findings in the field. For example, why replay switches directionality at different time points on a maze, how it can generate synthetic experiences, respects boundaries, etc. Being able to account for these different findings with a single mechanism is an important goal and no small feat.However, I still have some reservations about how much the current model differentiates itself – and represents an improvement – of a recently published model (the so-called prioritized memory access (PMA) model for replay Mattar and Daw (2018)). Further, although their model provides a relatively comprehensive account of multiple replay findings, it doesn't address many studies that show awake replay is non-local (even depicting experiences in remote environments). Finally, I would like the authors to elaborate on how their 'default' and 'reverse' policy could be implemented biologically and how these two modes relates to current thinking in the replay field. I elaborate on my points below:– Benefit over PMA model: The authors maintain that their model represents a significant improvement from the PMA model as the latter is not biologically feasible. In the PMA model, replay gets selected based on the utility of a possible experience reactivation which is partly derived from its "gain" term. Here, the authors' critique is that if the calculation of gain requires access to all stored experiences, this becomes very computationally laborious and biologically unfeasible. Yet, it's not clear to me in the current model's implementation that it represents a clear advantage in this way. Namely, to compute similarity – which influences replay prioritization – the current experience state needs to be compared with all other experience states (or tuples). Thus, does their model not also require access to all stored experiences? If not, I think the authors need to spell out a bit more about how this still makes their model more efficient and biologically implementable than the PMA model. This is a particularly important point as one of the main 'selling points' of the paper is that it achieves what the PMA model achieves but in a physiologically realistic way.

The reviewer is correct in stating that our model requires access to all experiences stored in memory – just like PMA does. To us, this is a reasonable assumption. The issue we are concerned about is the computation of the gain in PMA, which requires computing the change in the Q-function that would result, if a particular experience were reactivated. So, the brain would have to perform hypothetical updates to its network for each experience in memory, compute and store the gain for each experience, and then reactivate the experience with the highest gain. Since biological learning rules operate on synapses based on neural activity, we believe that it is unlikely that hypothetical updates can be computed and their outcome stored without altering the network. We clarified this point in the manuscript.

We added to the main text:

“Their model accounts for multiple observed replay phenomena. However, it is unclear how brains could compute or even approximate the computations required by Mattar and Daw's model. The brain would have to perform hypothetical updates to its network for each experience stored in memory, compute and store the utility for each experience, and then reactivate the experience with the highest utility. Since biological learning rules operate on synapses based on neural activity, it appears unlikely that hypothetical updates”.

– The authors show the model does well in replicating multiple replay findings in the field. However, they focus primarily on awake replay findings that show some relationship to current behaviour- e.g. depicting short cuts to goal locations, activating paths just taken, etc. However, many replay studies show that replay is very often non-local (e.g. Karlsson and Frank (2009); Olafsdottir et al. (2017)). How can their model accommodate these findings?

Our model can produce non-local awake replays if online replays are initialized in the same way as offline replays. We demonstrate this in additional simulation analysis in a new supplementary figure (Figure 2—figure supplement 2). We address the reviewer’s recommendation in more detail as part of our reply to the essential revision 1.

– In the current model there are two modes (policies) for computing experience similarity described; the default and reverse mode. The authors show you need both to effectively account for different replay phenomena (e.g. you need reverse for reward learning, particularly in environments where an animal's path is not prescribed). But what determines which policy is used? How would switching between the two be implemented in neural networks? Or do they propose they work in parallel?

Our model decides which of the modes should be used based on the recent (i.e., since that last trial) temporal difference errors. The probability of replay being generated using the reverse mode increases with the amount of temporal difference errors encountered. Neurons encoding reward prediction error have been known for (Schultz et al., 1997) and could be a trigger for the “switch” between replay modes. We address the reviewer’s recommendation in more detail as part of our reply to the essential revision 3.

– The default and reverse mode resonate with some contemporary theories of replay. Namely, a commonly held view is that awake replay may support planning/memory retrieval etc. whereas replay during rest periods supports memory consolidation. Perhaps the authors can elaborate on how their different modes, which resemble somewhat the planning vs consolidation mode, relate to these theories. Could it be that the default mode may be a sort of a planning mode – supporting on-going behaviour – whereas the reverse mode is more about learning?

We agree with reviewer and already discussed the different potential functions (including planning) of both replay modes in the discussion (“The Role of the Reverse and Default Modes”). However, we acknowledge that our phrasing might not have been clear enough. Hence, we rephrased parts of the discussion.

“Another function often attributed to hippocampal replay is that of planning (Diba and Buzsáki, 2007; Buckner, 2010; Buhry et al., 2011) by simulating possible future trajectories to inform subsequent behavior. The default mode could, in principle, support this function.”

Reviewer #3 (Recommendations for the authors):In multiple instances the authors refer to the biological plausibility of their model, and the lack thereof in the model by Mattar and Daw. Intuitively, I agree with the authors. However, to my knowledge we don't know what is biologically plausible and what is not. I suggest some rewording to emphasize that their model is simpler and less computationally taxing compared to Mattar and Daw. Further, I would appreciate a more explicit discussion of the biological plausibility of their model. They get into this to some degree in the discussion, particularly with regards to the return of inhibition term (i.e., line 401). However, the description of how experience strength or distance could be computed could certainly be expanded beyond just referencing other studies. Importantly, the model occurs in two modes. How might the hippocampus gate these two modes?

We agree with the reviewer that it is not exactly known what qualifies as biological plausible. Nonetheless, we would argue that the specific computations of PMA are unrealistic under the consideration of biological learning rules which operate on synapses based on neural activity. Namely, to compute the gain, the brain would have to perform hypothetical updates to its network for each experience in memory, compute and store the gain for each experience, and reactivate the experience with the highest gain. We find it unlikely that these computations can be performed without altering the network. We clarified this point in the manuscript and rephrased our criticism.

We extended the discussion of the biological plausibility of our model. Specifically, we more strongly relate experience similarity/distance to the activity of place cells which are theorized to encode just such a metric (Stachenfeld et al., 2017; Piray and Daw, 2021). Experience strength could be realized by place cell assemblies increasing their excitability in response to experience and reward. The replay mode switch could be triggered by neuron encoding reward prediction error (Schultz et al., 1997) and this is in part consistent with the increase of reverse replay following positive reward change (Ambrose et al., 2016, see our answer to Essential Revision 2). We address the reviewer’s recommendation in more detail as part of our reply to the essential revision 3.

In general I found the methods to be exquisitely articulated and comprehensive. However, the description of how replay is generated in their model could be expanded/clarified in the main text:– I understood that an experience consists of a transition between two states (in addition to reward/action)- and that the entire experience (including both states) is reactivated during replay. However I think a supplementary figure showing how exactly one gets from the last panel of figure 2A to an ordered set of reactivated states as in Figure 5c could be helpful.

We acknowledge that the generation of replay sequences might not be very intuitive. We added Figure 2—figure supplement 1 to show how a replay sequence is generated step-by-step.

– Most importantly- I think it would be very useful for the authors to show more examples of replay events generated by their model, particularly with respect to Figure 3. For example, they report the directionality of replays in the open field/labyrinth, but I would like to know where the replays actually started/ended, and how this changed over trials. This would help clarify how the replays are behaviorally useful to the agent.

We added Figure 3—figure supplement 2 with example replays from the replays that were used to produce Figure 3.

– The authors could better describe the logic/motivation of the default and reverse states. Initially I assumed that the default mode simply drives forward replay, and the reverse mode reverse replay- but this is not entirely the case. Why did the authors use just 2 modes (the first comparing the similarity between experience 1-state 1 and experience 2 state-1, and the second comparing the similarity between experience 1-state 1 and experience 2 state-2)- why not, for example, have other modes comparing experience 1-state 2 and experience 2 state-1, or experience 1-state 2 and experience 2 state-2? Many readers would benefit from a bit more intuition in terms of linking these modes to biological phenomenon.

The reviewer is correct that technically we could have simply stored sequences of experiences and replayed them in forward or reverse order. However, we wanted to develop and study a model for a simple mechanism that generates replay sequences. The reviewer is also correct that formally two more replay modes could be defined in our model. Let us call the other two cases “forward” mode (comparing experience 1-state 2 and experience 2 state-1) and “attractor” mode (comparing experience 1-state 2 and experience 2 state-2). These were not consistent with the goals of our study. In our paper, we wanted to study replay in our model that 1. generate sequences similar to the ones observed during SWR in the rodent hippocampus and 2. contribute in a specific way to learning. The attractor mode, and only this one mode, violates condition 1. since it looks for experiences that end in the same state, so that neighboring experiences are activated in a ring-like fashion (Figure 2—figure supplement 3). A number of previous studies have shown that reverse sequences are particularly efficient at propagation reward information backwards to decision points. So, condition 2 favors those modes (default and reverse) that consistently generate reverse sequences. The forward mode predominantly produces forward replays, which are not efficient for learning, but could very efficiently support planning – which is outside the scope of the current study. Note, that even though the default mode generally produces fewer reverse sequences than the reverse mode, in some situations it might predominantly generate reverse sequences. In addition, the default mode is more flexible in generating transitions between states that were never experienced, something that could be useful for discovering novel solutions.

In addition to adding Figure 2—figure supplement 3, we added a the above explanation to the results near l. 160.

– Can a single replay consist of both forward and reverse transitions? As mentioned above, a panel of example replays upfront could be very useful.

Indeed, this can happen, but this is limited to the default mode. Figure 3—figure supplement 2 shows some replays (during early learning phases) that exhibit this.

– Please confirm that only one map was constructed for linear environments. What was the reason for this choice, and would it impact the results to use two maps (one for each running direction, as in Mattar and Daw, and consistent with hippocampal fields on a linear track)?

We used only one map. If two maps were used and they were independent, the results would not be affected. If one-way transitions between the maps (current and next map) were allowed, then replay produced with the default mode at the end of a trial (end of the current map) could either be reverse (in the current map) or forward (in the next map). This would change the proportion of forward and reverse replay. However, this does not change any of the claims in this paper. That is why we used only one map for simplicity.

– A description of how the default versus reverse modes are selected in the dynamic mode is provided in the methods, but should also be described in the main text.

We added the description to the main text:

“Put simply, in the dynamic mode the probability of generating replay using the reverse mode increases with the TD errors accumulated since the last trial (for more details see Methods).”

– The definition of a replay epoch is in the methods but might be moved to the main text.

We moved the pseudo code summary of the replay epoch to the main text.

Can this model account for a broader scope of replay phenomena including: the sensitivity of reverse replay to reward magnitude or reward prediction error, over-representation of reward locations, or the tendency of forward replays to counter-intuitively lead towards an aversive (negatively rewarding) locations? As Mattar and Daw demonstrated that their model produces patterns of replay that match these arguably fundamental empirical findings, examining the fully overlapping set of scenarios would allow the models to be compared head-to-head. It is unclear whether either model can account for the recent observations that replay is enriched for locations not visited recently (Gillespie et al., 2021) or trajectories directed away from the animal's preferred goal location (Carey et al., 2019). I do not mean to suggest that the authors needs to perform all of the above tests. However, it would be of interest to at least include discussion about which empirical findings the current model could likely not account for.

We address the reviewer’s recommendation regarding the listed replay phenomena in our reply to the essential revision 2. The two recent studies mentioned by the reviewer (Carey et al., 2019. Gillespie et al., 2021) are indeed intriguing. We would agree with the reviewer that neither our model nor Mattar and Daw’s in their current form could account for these findings. A recent model presented by Antonov et al. (2022) which is based partly on PMA and includes “forgetting” in their learning process might account for both of them. We added a discussion on the studies by Carey et al. (2019) and Gillespie et al. (2021) as part of the section “Current Limitations of SFMA”. Furthermore, we also discuss how SFMA could be extended with a sort of recency prioritization to account for these two studies.

Line 160: 'in our simulations the reverse mode produces more updates that were optimal than did the default mode'. Please define an optimal (policy?) update.

The optimal update was defined here as the update with the highest “gain”. We rephrased the sentence to clarify what is meant by optimal:

“in our simulations the reverse mode produces more updates that were optimal, i.e., they had the highest utility according to Mattar and Daw (2018)’s model, than the default mode did”.

Line 165: I'm not sure what the authors mean by 'mixed' replays. Are they describing replays that traverse in one direction and then switch to the other (i.e., reverse to forward), as in Wu and Foster 2014?. If not, a better word choice might be 'interspersed'.

We changed the wording to ‘interspersed’ as suggested.

Line 188: Are there any statistical tests to accompany the statement that these log-log plots are linear?

At the current stage of this field, we are still focusing on qualitatively accounting for experimental observations, therefore quantitative comparisons between model and experimental data, as well as quantitative analyses of model data, are currently not constructive. Thus, we did not perform statistical tests to check for the linearity in the log-log plots.

Line 223: It is not clear to me what the alternating-alternating condition controls for.

We considered the alternating-alternating condition a kind of baseline in which the experience strengths of both maze sides is roughly balanced throughout the simulation. We changed the wording from ‘control’ to ‘baseline’.

Line 234: The explanation about the observed differences in short-cut replays under the different behavioral paradigms was somewhat confusing. A figure showing the relative amount of experience in each segment of the maze in each paradigm could help clarify.

This is a very good suggestion that we gladly followed (Figure 5—figure supplement 2).

Line 248: While it is interesting to ask whether the replay generated by your model facilitates learning in the task, I think it would be of broader interest and more informative to show specifically which replays are useful. For example, what was the distribution of replays that the default and reverse mode yielded on that task? How does task performance change if you limit replay to a subset, such as only reverse replays starting at the goal, or, most relevant here- only-short cut replays?

We fully agree with the reviewer that these questions are very interesting and important to answer. Some of these aspects, we had already address in the manuscript, e.g., the distribution of replays in default and reverse mode (Figure 3—figure supplement 1). However, we feel that further analysis of which subtypes of replay might drive learning differentially opens new questions that need to be addressed comprehensively in a separate manuscript.

Figure 6A: What are reactivation maps? I.e., is this a summary or average of all reactivations, or an example of an individual replay epoch (I assume it's the former, but that should be specified). Individual replay examples would also useful.

Indeed, the reactivation map is the average of all reactivations. We made the figure description clearer. We have also included more examples of individual replay events in Figure 2—figure supplement 1 and Figure 3—figure supplement 2.

Line 329: 'General representations of task structure in EC have been proposed to be forming hippocampal place cell code together with sensory information'- I suspect there was a typo in this sentence; I was unable to decipher its meaning.

We rephrased the sentence as:

“Hippocampal place cell code has been proposed to be formed by combining a learned representation of the task structure (e.g., space) with sensory information (Whittington et al., 2020).”.

Line 409: 'We argue that the minimum value should be applied after the summation of gain values" – what is the argument for applying the minimum value after summation? In general I wonder if the description of PMA's flaws (starting at line 404) should be moved to the related Supplemental Figure legend. I think more detail would be needed for most readers to understand the apparent error (for example, what is the 'special n-step update', what does it mean to apply before summation or after summation, and what makes one version better than the other)?

Our main criticism regarding the gain computation is that applying the minimum value artificially inflates the gain of sequences. We clarified the text regarding the n-step update and included parts of the text in the supplementary figure’s description (Figure 4—figure supplement 5).

Figure 5e: Please state in the figure legend what happened at trial 100. Also, as noted elsewhere, I think it would be useful to compare performance with the dynamic mode (and possibly the PMA model).

The reward shifted from left to right. We added clarification in the figure.

Figure 5/Supplementary Figure 7: I'm confused as to why in Figure 5 there is little difference between the forward and reverse modes in the initial learning of the task, but a difference after the reward location switch, whereas in Supplementary Figure 7 the exact opposite pattern is shown on the T-Maze. In general, what explains the differences in the initial learning rate and the learning rate after the reward switch, especially on the T-maze? I don't see what information the agent learned in the initial phase that remained useful after the switch. Perhaps showing examples of the agents behavioral trajectory/decisions at different stages of the task would help.

According to our modeling results two principles account for the difference pointed out by the reviewer: Reverse replay drives learning much more efficiently than random or forward replays and the default mode generates different fractions of forward and reverse replays depending on the statistics of the experiences. In Figure 5, choice was restricted to the decision point after which the agent would run straight forward until the next trial (running forward was imposed by the experimenter in Gupta et al. 2010). Hence, the task was quite simple and both replay modes lead to fast learning in the beginning. After the reward shift at trial 100 the agent has to unlearn the previously rewarded behavior. At the reward location, replay generated in the default mode can propagate either forward or in reverse order, whereas in the reverse mode sequences propagate only back towards the decision point. This makes the reverse mode perform better in unlearning the previously rewarded behavior in this specific task. For the simulations in the T maze shown in Figure S7 (now Figure 5—figure supplement 3), the agent can move freely and hence there is a difference in initial learning speeds between default and reverse mode, since the latter generates reverse sequences more reliably. However, when the rewarded side switches and the agent initially turns incorrectly, replay propagates backward in both replay modes because the agent is located at the end of the arm. So, the new reward information propagates with the same efficiency in both modes. Note, that in the open field (Figure 5—figure supplement 3, bottom) the default mode leads to lower learning speeds both during initial learning and after the reward is relocated, because behavior is less constraint and the default always generates fewer reverse sequences.

Supplementary Figure 3: what is the take-home message of showing the displacement distributions with different inverse temperature parameters? Is the point that the results hold when using different temperatures? If so, why not provide a more quantitative metric, like the log-log plots using different inverse temperature parameters?

The reviewer is partially correct in their assumption. The second motivation is to show that higher inverse temperature leads to less stochastic “diffusion” of replay sequences (Figure 4—figure supplement 2 B and D, as compared to A and C, respectively). The displacement distributions more intuitively demonstrates this than log-log plots would. We have clarified our motivation in the figure caption of Figure 4—figure supplement 2.

Supplementary Figure 8: The main text indicates that this figure was generated using the reverse mode, but the legend doesn't say so.

We changed the figure (Figure 6—figure supplement 1) to indicate that the reverse mode was used.

Methods line 463: 'The term in brackets is known as the temporal difference error'. This could perhaps be explained a bit more.

We added further explanation regarding the temporal difference error:

“The term in brackets is known as the temporal difference error, and represents by how much the last action and potentially collected reward has changed the expected discounted future reward associated with the state.”